# A MeA Tac1 neural circuit mediates anxiety-like behaviors in mice

Yao Wang[1,2], Jiu-Ye Qiao[1,2], Mei-Hui Yue[1,2], Xin-Yue Lv[1], Si-Ran Wang[1], Qian-Qian Yang[1], Han-Yun Kang[1], Hua-Li Yu[1], Xiao-Xiao He[1], Xiao-Juan Zhu [1✉] & Zi-Xuan He [1✉]

## Abstract

Anxiety is an emotion characterized by worried thoughts and feelings of unease, often accompanied by physical symptoms such as sweating and dizziness. Unlike other negative emotions, the neural circuits underlying anxiety are not well understood. Here we report that Tachykinin Precursor 1 (Tac1)-expressing neurons in the medial amygdala (MeA) respond to the transition from high anxiety to low anxiety states. The MeATac1 neurons regulate anxiety-like behaviors in mice bidirectionally. We also show that GABAergic neurons in the ventral tegmental area (VTA)$^{GABA}$→MeA$^{Tac1}$→ventrolateral part of the ventromedial hypothalamic nucleus (VMHvl) circuit contribute to anxiety-like behavior in mice. Our findings reveal a circuit of Tac1 neurons in the MeA that mediates anxiety-like behaviors in mice.

**Keywords** Anxiety; Medial Amygdala; Tachykinin Precursor 1; Ventral Tegmental Area; Ventrolateral Part of the Ventromedial Hypothalamic Nucleus
**Subject Category** Neuroscience

## Introduction

When the value judgment of an individual is negative, the individual experiences anxiety, which correspondingly triggers avoidance behaviors (Grupe and Nitschke, 2013; LeDoux, 2012). Anxiety plays a vital role in approach-avoidance behavior and survival in animals. A significant characteristic of anxiety is that it often cooccurs with many other negative emotions, such as depression and fear (Engin et al, 2016; He et al, 2019; He et al, 2020; Kalin, 2020; Tovote et al, 2015; Vink et al, 2008). This phenomenon suggests that the brain regions and neural circuits that are involved in the regulation of anxiety are complicated. Therefore, the investigation of the neural circuits in the brain that regulate anxiety is crucial for understanding brain function and the mechanisms underlying anxiety disorders.

Multiple regions of the brain are involved in the regulation of anxiety (Calhoon and Tye, 2015). Among these, the amygdala is a critical cluster of nuclei (Etkin and Wager, 2007; Grupe and Nitschke, 2013). The amygdala comprises multiple subdivisions, including the basolateral amygdala (BLA), MeA, and central amygdala (CeA). Research regarding the role of the amygdala in regulating anxiety has focused predominantly on the BLA and CeA (Felix-Ortiz et al, 2013; Huff et al, 2013; Tye et al, 2011); however, some studies have indicated that the MeA also plays a significant role in anxiety (Miller et al, 2019; Sotoudeh et al, 2022). The increased release of substance P into the MeA induces anxiety-like behavior in rats (Ebner et al, 2008; Ebner and Singewald, 2006). The precise mechanism by which the MeA is involved in anxiety-like behavior is currently poorly understood. The identification of the neural circuits in the MeA that are involved in the regulation of anxiety can enhance our understanding of the mechanisms in the brain underlying anxiety-related behavior.

The VTA is a key region for reward and aversion (Hu, 2016). However, the role of VTA function in anxiety is not yet fully understood. The impairment of VTA neuron function leads to anxiety-like behaviors (Zweifel et al, 2011), and the activation of VTA neurons also induces anxiety in animals (Qi et al, 2022). These inconsistent results may be attributable to differences in the regions downstream of VTA pathways. The VMHvl is a core brain area involved in aggressive behavior (Lee et al, 2014; Lin et al, 2011), and it has also been found to modulate anxiety-like behaviors. An increase in VMHvl burst firing has been observed after chronic stress, and the inhibition of burst firing has been shown to rescue anxiety-like behaviors (Shao et al, 2022). Steroidogenic factor-1 (SF-1)-positive neurons in the VMHvl regulate anxiety levels in mice, and the ablation of SF-1 neurons in the VMHvl reduces the level of anxiety in mice (Cheung et al, 2015; Kunwar et al, 2015).

Herein, we showed that increased MeA$^{Tac1}$ neuronal activity is tightly coupled with the transition from a high anxiety level to a low anxiety level. MeA$^{Tac1}$ neurons regulate anxiety-like behaviors in mice. Moreover, chemogenetic manipulation of the VTA$^{GABA}$→MeA$^{Tac1}$→VMHvl circuit bidirectionally mediates anxiety-like behaviors. These results demonstrate the crucial role of the VTA$^{GABA}$→MeA$^{Tac1}$→VMHvl circuit in regulating anxiety.

[1]Key Laboratory of Molecular Epigenetics, Ministry of Education, Institute of Genetics and Cytology, Northeast Normal University, 130021 Changchun, China. [2]These authors contributed equally: Yao Wang, Jiu-Ye Qiao, Mei-Hui Yue. ✉E-mail: zhuxj720@nenu.edu.cn; hezx234@nenu.edu.cn

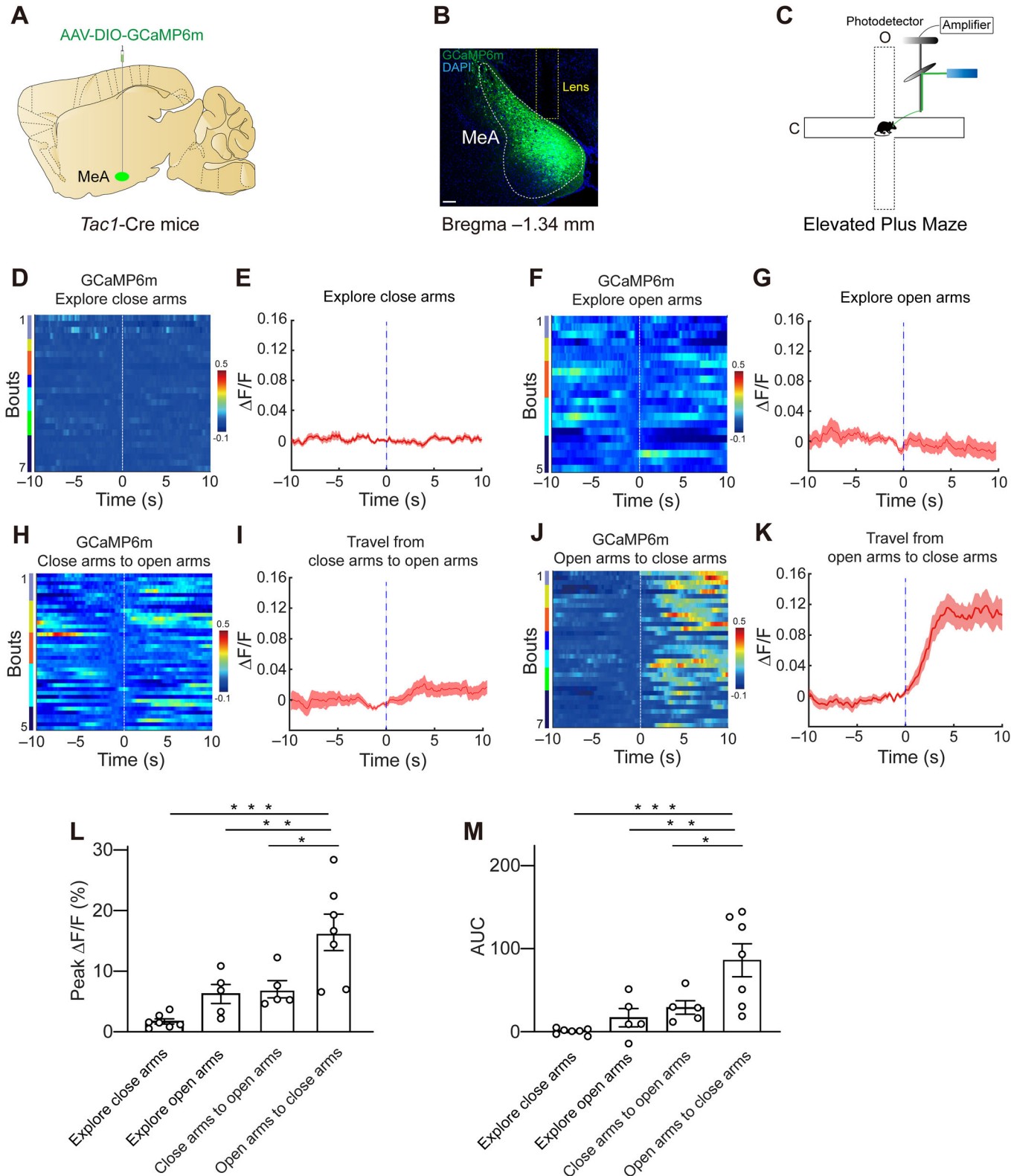

**Figure 1. Ca²⁺ activity of MeA^Tac1 neurons during anxiety-like behaviors.**

(A, B) Images are representative of fiber tracks and GCaMP6m expression in the MeA. Scale bar, 100 μm. (C) Schematic illustration of the fiber photometry. (D) Heatmap of Ca²⁺ signals from MeA^Tac1 neurons exploring closed arms. Signals are aligned to the time point when mice explore through the closed arms. In the heatmap, the different color bars on the left represent different individual mice. $n = 26$ trials from seven mice. (E) Peri-event plot of ΔF/F Ca²⁺ signal changes in MeA^Tac1 neurons exploring closed arms. (F) Heatmap of Ca²⁺ signals from MeA^Tac1 neurons exploring open arms. Signals are aligned to the time point when mice explore through the open arms. In the heatmap, the different color bars on the left represent different individual mice. $n = 21$ trials from five mice. (G) Peri-event plot of ΔF/F Ca²⁺ signal changes in MeA^Tac1 neurons exploring open arms. (H) Heatmap of Ca²⁺ signals from MeA^Tac1 neurons during the transition from the closed arms to the open arms. Signals are aligned to the time point when mice move from the closed arms to the open arms. In the heatmap, the different color bars on the left represent different individual mice. The time "0" of the coordinate axis refers to the moment the mouse transitions from the closed arms to the open arms. $n = 40$ trials from five mice. (I) Peri-event plot of ΔF/F Ca²⁺ signal changes in MeA^Tac1 neurons during the transition from the closed arms to the open arms. (J) Heatmap of Ca²⁺ signals from MeA^Tac1 neurons during the transition from the open arms to the closed arms. Signals are aligned to the time point when mice move from the open arms to the closed arms. In the heatmap, the different color bars on the left represent different individual mice. The time "0" of the coordinate axis refers to the moment the mouse transitions from the open arms to the closed arms. $n = 34$ trials from 7 mice. (K) Peri-event plot of ΔF/F Ca²⁺ signal changes in MeA^Tac1 neurons during the transition from the open arms to the closed arms. (L) Peak Ca²⁺ signals. exploring closed arms: $N = 7$; exploring open arms: $N = 5$; transition from the closed arms to the open arms: $N = 5$; transition from the open arms to the closed arms: $N = 7$. (M) Area under the curve (AUC). exploring closed arms: $N = 7$; exploring open arms: $N = 5$; transition from the closed arms to the open arms: $N = 5$; transition from the open arms to the closed arms: $N = 7$. $n$ trial number, $N$ animal number, MeA medial amygdala. All of the data are presented as the means ± s.e.m.s. See Table EV1 for detailed statistics. Source data are available online for this figure.

# Results

## Activity of MeA^Tac1 neurons during anxiety-like behaviors

To investigate whether MeA^Tac1 neurons are involved in regulating anxiety-like behaviors in mice, we conducted fiber photometry experiments. First, we injected an adeno-associated virus (AAV) expressing the GCaMP6m virus into the MeA of Tac1-Cre mice (Fig. 1A,B). In the elevated plus maze test, mice experienced increased anxiety when they explored the open arms, which lack walls on both sides. Conversely, when mice explored the closed arms, their anxiety level decreased (Fig. 1C). Therefore, by monitoring the activity of MeA^Tac1 neurons during the transition from an open arm to a closed arm in the EPM test, we aimed to investigate the role of MeA^Tac1 neurons in anxiety.

Upon analyzing the calcium imaging results for mice in which the virus was successfully injected into the MeA, we observed that during the transition from an open arm to a closed arm, the activity of MeA^Tac1 neurons increased (Fig. 1J,K). However, we failed to observe a change in the activity of MeA^Tac1 neurons when the mice were transferred from a closed arm to an open arm (Fig. 1H,I). Additionally, there was no significant change in the activity of MeA^Tac1 neurons when the mice explored the closed or open arms (Fig. 1D–G). These findings suggest that when mice experienced a reduction in anxiety levels, the activity of Tac1 neurons specifically increased, thus suggesting that MeA^Tac1 neurons may play an important role in regulating anxiety-like behaviors in mice (Fig. 1L,M).

## The activity of MeA^Tac1 neurons plays a crucial role in anxiety

During the exploration of the elevated plus maze, when the mice move from the open arms to the closed arms, leading to a decrease in anxiety levels, the activity of Tac1 neurons increases concurrently. As previous work demonstrates (Jimenez et al, 2018), this suggests that Tac1 neurons in the MeA may play an important role in regulating anxiety-like behavior in mice.

To determine whether MeA^Tac1 neurons regulate anxiety, we chemogenetically manipulated their activity by using designer receptors that are exclusively activated by designer drugs (DREADDs). To selectively manipulate the activity of MeA^Tac1 neurons, we bilaterally injected AAV-DIO-hM3Dq-mCherry and AAV-DIO-mCherry into the MeA of Tac1-Cre mice (Fig. 2A,B). We subsequently examined whether the activation of MeA^Tac1 neurons regulated anxiety-like behaviors in mice. We then conducted the following three behavioral assays to measure anxiety-like behaviors in the mice: the open field test (OFT), the elevated plus maze (EPM) test, and the light–dark box (LDB) test. Compared with the distance traveled in the control group (mice injected with mCherry), the total distance traveled in the arena of the mice in the chemogenetic activation group (mice injected with hM3Dq) did not differ. However, hM3Dq mice spent significantly more time exploring the center area of the arena (Fig. 2C–E), thus indicating that the activation of MeA^Tac1 neurons suppressed anxiety-like behaviors. In the EPM test, increased activity of MeA^Tac1 neurons led to increased time spent exploring the open arms (Fig. 2F,G). Similarly, in the LDB assay, hM3Dq mice spent significantly more time in the light chamber (Fig. 2H). These results demonstrate that an increase in the activity of Tac1 neurons inhibits anxiety-like behaviors in mice.

We further assessed whether the inhibition of the activity of MeA^Tac1 neurons promotes anxiety-like behaviors in mice. We bilaterally injected AAV-DIO-hM4Di-mCherry and AAV-DIO-mCherry into the MeA of Tac1-Cre mice (Fig. 2I,J). Compared with control mice, hM4Di-injected mice spent less time exploring the center area of the arena in the OFT, thus indicating that the inhibition of MeA^Tac1 neurons promoted anxiety-like behaviors (Fig. 2K–M). In the EPM test, decreased activity of MeA^Tac1 neurons led to less time spent in the open arms (Fig. 2N,O). Furthermore, in the LDB test, hM4Di-injected mice spent significantly less time in the light chamber (Fig. 2P). In the tail suspension and sucrose preference tests, the activity of MeA^Tac1 neurons was not required for depression-like behaviors in the mice (Fig. EV1A,B). When considering the potential impact of sex on the behavioral results, we repeated the experiments in female mice and found that the inhibition of MeA^Tac1 neurons also induced anxiety-like behaviors in female mice (Fig. EV2). In summary, our data suggest that MeA^Tac1 neurons play a crucial role in modulating anxiety-like behaviors in mice.

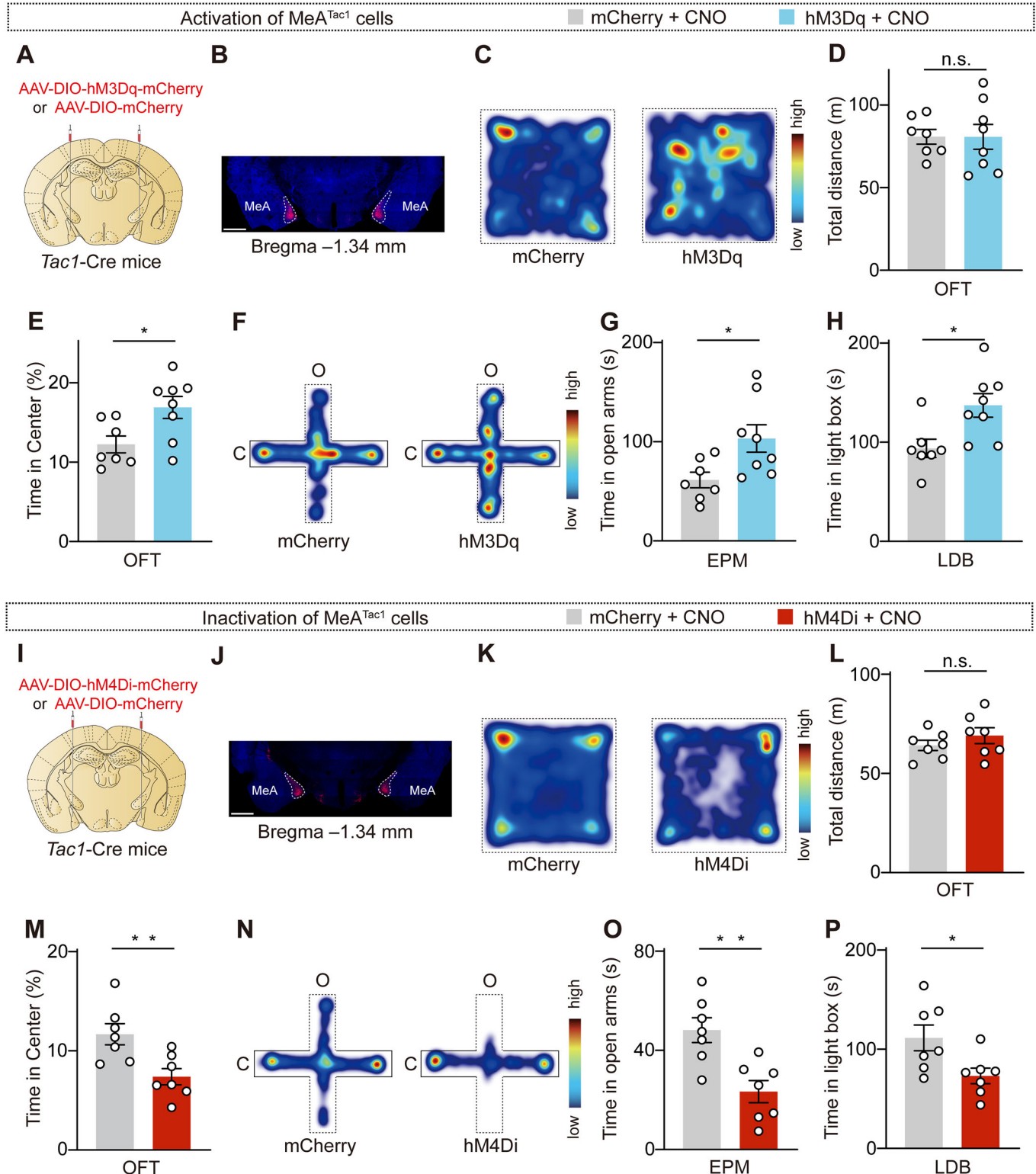

## Activity of GABA neurons in the VTA during anxiety-like behaviors

Subsequently, we sought to identify the upstream brain regions with inputs to Tac1 neurons that are potentially involved in regulating anxiety-like behaviors in mice. We utilized a monosynaptic viral tracing strategy in *Tac1*-Cre mice. We first injected AAV-DIO-RVG or AAV-DIO-TVA-GFP into the MeA. Two weeks later, we injected RV-EnVA-dsRed at the same coordinates (Fig. EV3A,B). Our results revealed several upstream brain regions

**Figure 2.  MeA$^{Tac1}$ neurons mediate anxiety-like behaviors in mice.**

(A) Schematic of the strategies that were used to express AAV-DIO-mCherry and AAV-DIO-hM3Dq-mCherry in the MeA of *Tac1*-Cre mice. (B) Representative images showing virus expression in the MeA. Scale bar, 1 mm. (C) Heatmaps displaying the time spent in different regions of the open field arena (warmer colors indicate more time spent in a region). (D) Total distance traveled in the open field arena. $P = 0.9962$. mCherry: $N = 7$; hM3Dq: $N = 8$. (E) Increased time spent in the center of the open field arena was observed in hM3Dq mice. $P = 0.0222$. mCherry: $N = 7$; hM3Dq: $N = 8$. (F) Heatmaps displaying the time spent in different regions of the elevated plus maze (warmer colors indicate more time spent in a region). (G) Increased time spent in the open arms was observed in hM3Dq mice. $P = 0.0255$. mCherry: $N = 7$; hM3Dq: $N = 8$. (H) Increased time spent in the light box was observed in hM3Dq mice. $P = 0.0145$. mCherry: $N = 7$; hM3Dq: $N = 8$. (I) Schematic of the strategies that were used to express AAV-DIO-mCherry and AAV-DIO-hM4Di-mCherry in the MeA of *Tac1*-Cre mice. (J) Representative images showing virus expression in the MeA. Scale bar, 1 mm. (K) Heatmaps displaying the time spent in different regions of the open field arena (warmer colors indicate more time spent in a region). (L) Total distance traveled in the open field arena. $P = 0.3238$. mCherry: $N = 7$; hM4Di: $N = 7$. (M) Decreased time spent in the center of the open field arena was observed in hM4Di mice. $P = 0.0078$. mCherry: $N = 7$; hM4Di: $N = 7$. (N) Heatmaps displaying the time spent in different regions of the elevated plus maze (warmer colors indicate more time spent in a region). (O) Decreased time spent in the open arms was observed in hM4Di mice. $P = 0.0030$. mCherry: $N = 7$; hM4Di: $N = 7$. (P) Decreased time spent in the light box was observed in hM4Di mice. $P = 0.0266$. mCherry: $N = 7$; hM4Di: $N = 7$. $N$ animal number. MeA: medial amygdala. Statistical significance was determined via an unpaired $t$ test. All of the data are presented as the means ± s.e.m.s. *$P < 0.05$; **$P < 0.01$; n.s. not significant. See Table EV1 for detailed statistics. Source data are available online for this figure.

with inputs to the MeA, including the prelimbic cortex (PrL), posteromedial cortical amygdaloid area (PMCo), posterolateral cortical amygdaloid area (PLCo), ventral tegmental area (VTA), and dorsal paragigantocellular nucleus (DPGi) (Fig. EV3C,D). The VTA is a brain region that is known to play a crucial role in regulating anxiety (Calhoon and Tye, 2015; Tovote et al, 2015). In subsequent experiments, we performed immunofluorescence experiments in the VTA and found that these RV-positive neurons expressed the GABAergic neuron marker GAD67 but did not express the excitatory neuron marker VGLUT2 or the dopaminergic neuron marker TH (Fig. EV3E).

Upon analyzing the calcium imaging results for the mice in which the virus was successfully injected into the GABA neurons in the VTA (Fig. 3A–C), we observed that during the transition from a closed arm to an open arm, the activity of VTA$^{GABA}$ neurons increased (Fig. 3H,I). Moreover, we observed decreased activity of VTA$^{GABA}$ neurons when the mice transferred from an open arm to a closed arm (Fig. 3J,K). However, there was no significant change in the activity of VTA$^{GABA}$ neurons when the mice explored the closed arms or open arms (Fig. 3D–G). These findings suggest that when mice experience changes in anxiety levels, the activity of GABA neurons in the VTA, which project to MeA$^{Tac1}$, specifically changes, thus suggesting that VTA$^{GABA}$ neurons may play an important role in regulating anxiety-like behaviors in mice (Fig. 3L,M).

### The VTA$^{GABA}$→MeA$^{Tac1}$ pathway regulates anxiety

To investigate whether projections from the VTA to MeA$^{Tac1}$ neurons are functionally involved in anxiety-like behaviors in mice, we examined whether the activation of VTA→MeA$^{Tac1}$ projections could alleviate anxiety. We bilaterally injected AAV-fDIO-mCherry (control) or AAV-fDIO-hM3Dq-mCherry into the MeA and injected the anterograde virus AAV1-DIO-flpo into the VTA of *Tac1*-Cre mice (Fig. 4A,B). After 4 weeks, we conducted behavioral assays to measure anxiety. In the OFT, we observed a significant increase in the time spent in the center of the open field arena by hM3Dq-injected mice following the administration of CNO (Fig. 4C–E). Additionally, in the EPM test, hM3Dq-injected mice spent more time exploring the open arms (Fig. 4F,G), thus indicating reduced anxiety-like behaviors. Furthermore, in the LDB test, hM3Dq-injected mice spent more time exploring the light box (Fig. 4H). These findings suggest that the activation of the VTA→MeA$^{Tac1}$ projections ameliorates anxiety-like behaviors in

mice. In the following experiments, C57BL/6J mice were bilaterally injected with AAV-DIO-mCherry (control) or AAV-DIO-hM4Di-mCherry virus into the VTA and a retrograde AAV-Retro-CAG-Cre virus into the MeA (Fig. EV4A,B). We found that inactivation of MeA-projecting VTA neuron activity through CNO (5 µM in 0.3 µl) treatment in C57BL/6 J mice inhibited anxiety-like behaviors in mice (Fig. EV4C–H).

We further investigated whether the inactivation of VTA→MeA$^{Tac1}$ projections inhibits anxiety. *Tac1*-Cre mice received bilateral injections of AAV-fDIO-mCherry (control) or AAV-fDIO-hM4Di-mCherry into the MeA and an injection of the anterograde virus AAV1-DIO-flpo into the VTA (Fig. 4I,J). After 4 weeks, we conducted behavioral assays. Compared with control animals, hM4Di-injected mice spent significantly less time in the center of the open field arena after the administration of CNO (Fig. 4K–M). Subsequent experiments demonstrated that the activation of the VTA→MeA$^{Tac1}$ projections led to reduced exploration time in the open arms and light box by hM4Di-injected mice (Fig. 4N–P). In the following experiments, C57BL/6J mice were bilaterally injected with AAV-DIO-mCherry (control) or AAV-DIO-hM3Dq-mCherry virus into the VTA and a retrograde AAV-Retro-CAG-Cre virus into the MeA (Fig. EV4I,J). We found that activation of MeA-projecting VTA neuron activity through CNO (5 µM in 0.3 µl) treatment in C57BL/6J mice promoted anxiety-like behaviors in mice (Fig. EV4K–P). Taken together, these findings suggest that VTA→MeA$^{Tac1}$ projections play a regulatory role in anxiety.

### The MeA$^{Tac1}$→VMHvl pathway modulates anxiety

In our previous study, we reported that MeA$^{Tac1}$ neurons primarily project to the VMHvl (He et al, 2024). In subsequent experiments, to determine whether MeA$^{Tac1}$→VMHvl projections play a functional role in anxiety, we first assessed whether the activation of MeA$^{Tac1}$→VMHvl projections inhibited anxiety. *Tac1*-Cre mice were bilaterally injected with AAV-fDIO-mCherry (control) or AAV-fDIO-hM3Dq-mCherry into the MeA and with the retrograde virus AAV-Retro-FLEX-flpo into the VMHvl (Fig. 5A,B). Four weeks later, we performed behavioral assays to evaluate anxiety. Compared with control animals, hM3Dq-injected mice spent significantly more time in the center of the arena following CNO administration (Fig. 5C–E). In the EPM test, hM3Dq-injected mice spent more time exploring the open arms than control mice (Fig. 5F,G). Moreover, in the LDB test, the activation of MeA$^{Tac1}$

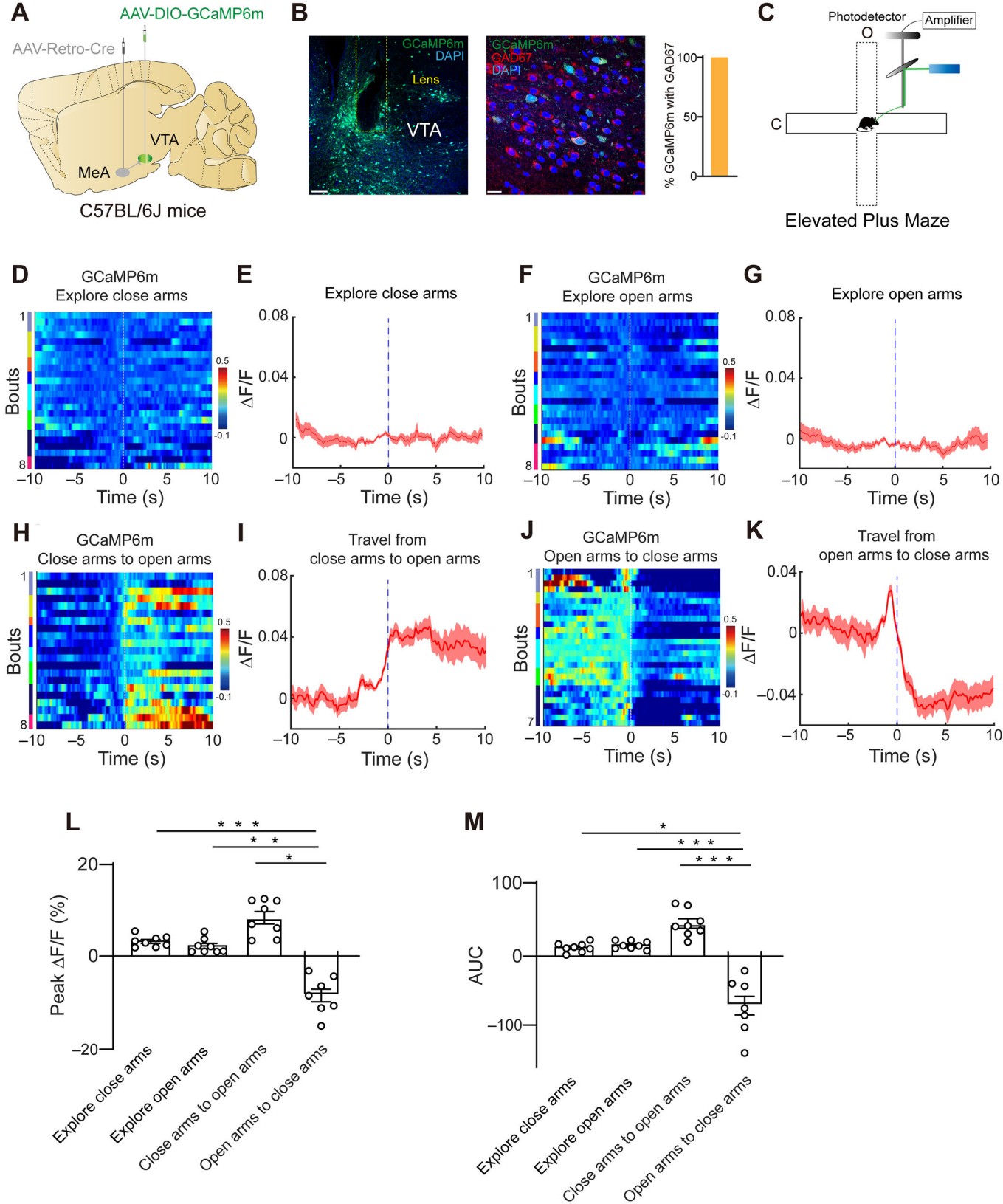

◄ **Figure 3. Ca²⁺ activity of VTA^GABA neurons during anxiety-like behaviors.**

(A, B) Images are representative of fiber tracks and GCaMP6m expression in the VTA. Scale bar, 100 μm and 20 μm. Colocalization analysis was performed on 87 cells obtained from 9 tissue slices across 3 mice. (C) Schematic illustration of the fiber photometry. (D) Heatmap of Ca²⁺ signals from VTA^GABA neurons exploring closed arms. Signals are aligned to the time point when mice explore through the closed arms. In the heatmap, the different color bars on the left represent different individual mice. $n = 24$ trials from eight mice. (E) Peri-event plot of ΔF/F Ca²⁺ signal changes in VTA^GABA neurons exploring closed arms. (F) Heatmap of Ca²⁺ signals from VTA^GABA neurons exploring open arms. Signals are aligned to the time point when mice explore through the open arms. In the heatmap, the different color bars on the left represent different individual mice. $n = 24$ trials from eight mice. (G) Peri-event plot of ΔF/F Ca²⁺ signal changes in VTA^GABA neurons exploring open arms. (H) Heatmap of Ca²⁺ signals from VTA^GABA neurons during the transition from the closed arms to the open arms. Signals are aligned to the time point when mice move from the closed arms to the open arms. In the heatmap, the different color bars on the left represent different individual mice. The time "0" of the coordinate axis refers to the moment the mouse transitions from the closed arms to the open arms. $n = 21$ trials from eight mice. (I) Peri-event plot of ΔF/F Ca²⁺ signal changes in VTA^GABA neurons during the transition from the closed arms to the open arms. (J) Heatmap of Ca²⁺ signals from VTA^GABA neurons during the transition from the open arms to the closed arms. Signals are aligned to the time point when mice move from the open arms to the closed arms. In the heatmap, the different color bars on the left represent different individual mice. The time "0" of the coordinate axis refers to the moment the mouse transitions from the open arms to the closed arms. $n = 27$ trials from seven mice. (K) Peri-event plot of ΔF/F Ca²⁺ signal changes in VTA^GABA neurons during the transition from the open arms to the closed arms. (L) Peak Ca²⁺ signals. (M) Area under the curve (AUC). $n$ trial number, $N$ animal number. All of the data are presented as the means ± s.e.m.s. See the Table EV1 for detailed statistics. Source data are available online for this figure.

neurons led to an increase in the time spent exploring the light box (Fig. 5H). In summary, the activation of the MeA^Tac1→VMHvl neuronal pathway via CNO treatment attenuates anxiety-like behaviors in mice.

We next investigated whether the inactivation of MeA^Tac1→VMHvl projections promotes anxiety. *Tac1*-Cre mice were bilaterally injected with AAV-fDIO-mCherry (control) or AAV-fDIO-hM4Di-mCherry into the MeA and with the retrograde virus AAV-Retro-FLEX-flpo into the VMHvl (Fig. 5I,J). In the OFT, the administration of CNO induced a marked decrease in the time spent in the center area (Fig. 5K–M). In the EPM and LDB tests, the time spent exploring the open arms and light box by the hM4Di-injected mice was reduced (Fig. 5N–P). Taken together, these data indicate that MeA^Tac1→VMHvl projections regulate anxiety.

## The VTA^GABA→MeA^Tac1→VMHvl circuit mediates anxiety

In our previous study, we reported that MeA^Tac1 neurons are GABAergic (He et al, 2024). To characterize the functional connectivity between MeA^Tac1 neurons and the VMHvl, we injected a ChR2-expressing AAV into the MeA of Tac1-Cre mice. Upon optogenetic stimulation of MeA axonal terminals in the VMHvl, we observed light-induced inhibitory postsynaptic currents (IPSCs) in VMHvl neurons. These IPSCs were entirely abolished by the voltage-gated sodium channel blocker tetrodotoxin (TTX) but were partially restored by subsequent application of the voltage-gated potassium channel blocker 4-aminopyridine (4-AP) following TTX application. The complete blockade of IPSCs was achieved by the administration of the NMDA receptor antagonist APV and the GABA_A receptor antagonist bicuculline (Fig. EV5A–C). These findings suggest a direct monosynaptic GABAergic connection between the MeA and VMHvl.

We next investigated whether the VTA^GABA→MeA^Tac1→VMHvl circuit regulates anxiety through a combination of region-specific chemogenetic manipulations. We unilaterally injected AAV-fDIO-hM4Di-mCherry into the MeA and the anterograde virus AAV1-DIO-flpo into the VTA of *Tac1*-Cre mice. A cannula was then implanted into the VMHvl, through which CNO (5 μM in 0.3 μl) or saline was infused (Fig. 6A,B). After 4 weeks, behavioral assays were conducted to assess anxiety. CNO administration resulted in a significant reduction in the time spent exploring the center of the arena in the OFT (Fig. 6C–E). Similarly, in the EPM and LDB tests, mice infused with CNO spent less time in the open arms and light box (Fig. 6F–H).

Substance P (SP) serves as a neurotransmitter and is encoded by tachykinin 1 (*Tac1*), which produces SP and neurokinin A via alternative slicing and posttranslational modifications (Krause et al, 1987; Levine et al, 1993; Nicoll et al, 1980; Otsuka and Yoshioka, 1993). In subsequent experiments, to investigate whether the release of SP from MeA^Tac1 neurons is crucial for anxiety-like behaviors in mice, AAV-TAC1-shRNA or AAV-scramble-shRNA was used in the MeA (Fig. EV6A). Quantitative RT-PCR confirmed the knockdown efficiency (Fig. EV6B). The results of the anxiety behavioral assays revealed that *Tac1* knockdown in the MeA failed to induce anxiety-like behaviors in the mice (Fig. EV6C–H). These results suggest that the release of GABA from MeA^Tac1 neurons may play an important role in anxiety. Subsequently, the GABA_A receptor antagonist bicuculline (50 μM in 0.3 μl) was administered into the VMHvl via cannula infusion (Fig. EV7A,B). Our results demonstrated that the introduction of bicuculline leads to anxiety-like behaviors in mice (Fig. EV7C–H). Upon further inspection, we investigated whether the VTA^GABA→MeA^Tac1→VMHvl circuit modulates anxiety through a combination of region-specific pharmacological and chemogenetic manipulations. We unilaterally injected AAV-fDIO-hM3Dq-mCherry into the MeA and the anterograde virus AAV1-DIO-flpo into the VTA of *Tac1*-Cre mice. A cannula was then implanted into the VMHvl, through which bicuculline or saline was infused (Fig. 6I,J). After 4 weeks, behavioral assays were conducted to assess anxiety. Bicuculline administration resulted in a significant reduction in the time spent exploring the center of the arena in the OFT (Fig. 6K–M). Similarly, in the EPM and LDB tests, mice infused with bicuculline spent less time in the open arms and light box (Fig. 6N–P). In summary, the VTA^GABA→MeA^Tac1→VMHvl circuit plays a crucial role in modulating anxiety-like behaviors in mice.

## Discussion

In our previous study, we reported that Tac1 neurons in the MeA project primarily to the downstream VMHvl; at this location, substance P (which is secreted by Tac1 neurons) regulates aggressive behaviors in mice (He et al, 2024). In this study, we found that although Tac1 neurons secrete both substance P and GABA, substance P does not affect anxiety-like behavior in mice. Instead, the secretion of GABA to the downstream VMHvl modulates anxiety-like behaviors in mice. These findings suggest that the neural circuits related to anxiety are closely linked and interact with those related to other negative

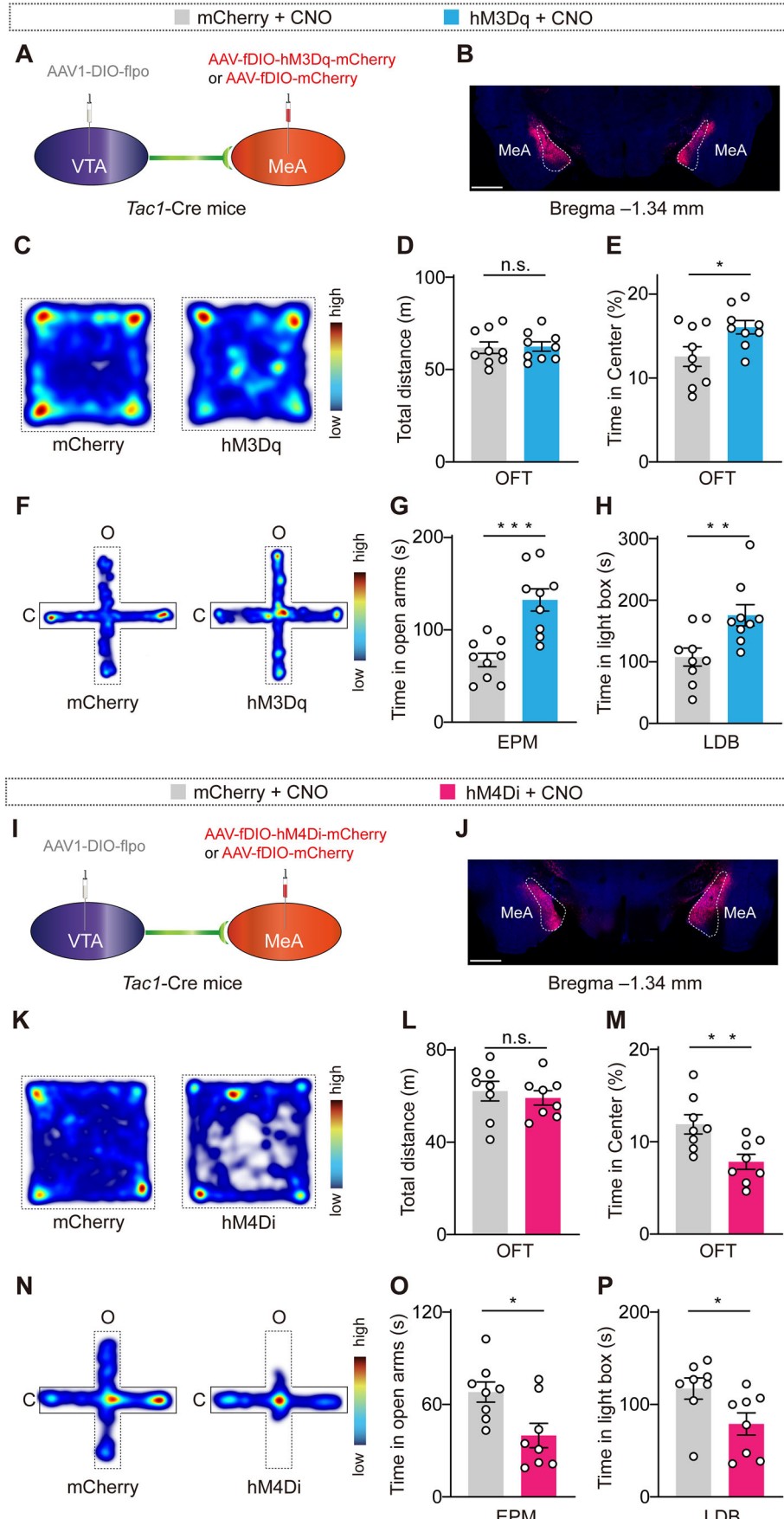

◄ **Figure 4. The VTA→MeA^Tac1 pathway regulates anxiety-like behaviors in male mice.**

(A) Schematic of the strategies that were used to express AAV1-DIO-flpo in the VTA and AAV-fDIO-mCherry or AAV-fDIO-hM3Dq-mCherry in the MeA of *Tac1*-Cre mice. (B) Representative images showing virus expression in the MeA. Scale bar, 1 mm. (C) Heatmaps displaying the time spent in different regions of the open field arena (warmer colors indicate more time spent in a region). (D) Total distance traveled in the open field arena. $P = 0.8770$. mCherry: $N = 9$; hM3Dq: $N = 9$. (E) Time spent in the center of the arena. $P = 0.0254$. mCherry: $N = 9$; hM3Dq: $N = 9$. (F) Heatmaps displaying the time spent in different regions of the elevated plus maze (warmer colors indicate more time spent in a region). (G) Time spent in the open arms. $P = 0.0003$. mCherry: $N = 9$; hM3Dq: $N = 9$. (H) Time spent in the light box. $P = 0.0085$. mCherry: $N = 9$; hM3Dq: $N = 9$. (I) Schematic of the strategies that were used to express AAV1-DIO-flpo in the VTAI and AAV-fDIO-mCherry or AAV-fDIO-hM3Dq-mCherry in the MeA of *Tac1*-Cre mice. (J) Representative images showing virus expression in the MeA. Scale bar, 1 mm. (K) Heatmaps displaying the time spent in different regions of the open field arena (warmer colors indicate more time spent in a region). (L) Total distance traveled in the open field arena. $P = 0.5813$. mCherry: $N = 8$; hM4Di: $N = 8$. (M) Time spent in the center of the arena. $P = 0.0081$. mCherry: $N = 8$; hM4Di: $N = 8$. (N) Heatmaps displaying the time spent in different regions of the elevated plus maze (warmer colors indicate more time spent in a region). (O) Time spent in the open arms. $P = 0.0156$. mCherry: $N = 8$; hM4Di: $N = 8$. (P) Time spent in the light box. $P = 0.0371$. mCherry: $N = 8$; hM4Di: $N = 8$. $N$ animal number, MeA medial amygdala. Statistical significance was determined via an unpaired $t$ test. All of the data are presented as the means ± s.e.m.s. *$P < 0.05$; **$P < 0.01$; ***$P < 0.001$; n.s., not significant. See Table EV1 for detailed statistics. Source data are available online for this figure.

emotions, which may help to explain why anxiety often cooccurs with other neurological disorders.

The VTA^GABA neurons are believed to play a critical regulatory role in anxiety (Jennings et al, 2013; McHenry et al, 2017; Nieh et al, 2016). Our findings indicate that when mice move from a closed arm to an open arm, their anxiety levels increase, thus leading to increased activity in VTA^GABA neurons. This results in stronger inhibition of their downstream projection neurons. Conversely, when anxiety levels decrease, the activity of VTA^GABA neurons diminishes, thus leading to reduced inhibition of downstream neurons. These results are consistent with our findings in MeA^Tac1 neurons. The increased activity of these neurons can alleviate anxiety-like behavior in mice, whereas reduced activity can intensify such behavior. However, we cannot rule out the possibility that projections from other upstream brain regions to VTA^GABA neurons also participate in regulating anxiety-like behavior.

We observed that the introduction of GABA receptor antagonist, bicuculline into the downstream VMHvl region suppressed anxiety-like behaviors in mice. These findings suggest that GABA released from MeA^Tac1 neurons may play a significant regulatory role in modulating anxiety-like behaviors. However, due to the limitations of our methodology, we cannot rule out the possibility that GABA released from other brain regions project to the VMHvl also contributes to the regulation of anxiety-like behaviors in mice.

In summary, the VTA^GABA→MeA^Tac1→VMHvl pathway is a crucial component of the circuit that controls anxiety-like behaviors in mice. These results may contribute to the understanding of anxiety-related circuits.

## Methods

### Reagents and tools table

| Reagent/resource | Reference or source | Identifier or catalog number |
|---|---|---|
| **Experimental models** | | |
| Tac1-IRES2-Cre mice | The Jackson Laboratory | 021877 |
| C57BL/6J mice | The Yisi Animal Center | N/A |
| **Virus** | | |
| AAV-EF1α-DIO-EGFP-T2A-TVA | Brain Case | BC-0041 |
| AAV-EF1α-DIO-oRVG | Brain Case | BC-0442 |

| Reagent/resource | Reference or source | Identifier or catalog number |
|---|---|---|
| RV-ENVA-ΔG-mCherry | Brain Case | BC-RV-EnvA844 |
| AVV-CAG-Retro-Cre | Brain Case | BC-0161 |
| AAV-EF1α-DIO-GCaMP6m | Brain Case | BC-0087 |
| AAV-EF1α-DIO-hM4D(Gi)-mCherry | Brain Case | BC-0155 |
| AAV-EF1α-DIO-hM3D(Gq)-mCherry | Brain Case | BC-0146 |
| AAV-EF1α-DIO-mCherry | Brain Case | BC-0016 |
| AAV-EF1α-fDIO-hM4D(Gi)-mCherry | Brain Case | BC-1232 |
| AAV-EF1α-fDIO-hM3D(Gq)-mCherry | Brain Case | BC-0495 |
| AAV-EF1α-fDIO-mCherry | Brain Case | BC-0177 |
| AAV-Retro-CAG-FLEX-flpo | Brain Case | BC-0176 |
| AAV-EF1α-DIO-flpo | Brain Case | BC-0178 |
| AAV-EF1α-DIO-ChR2-mCherry | Brain Case | BC-0103 |
| AAV-scramble-shRNA-mCherry | Brain Case | N/A |
| AAV-TAC1-shRNA-mCherry | Brain Case | N/A |
| **Antibodies** | | |
| Rabbit anti-VLGUT2 | Synaptic Systems | 135 402 |
| Rat anti-GAD67 | Abcam | ab75712 |
| Rabbit anti-Tyrosine Hydroxylase | Abcam | ab6211 |
| Alexa Fluor 488-conjugated goat anti-rabbit | Invitrogen | A11008 |
| **Chemicals, enzymes and other reagents** | | |
| CNQX | Tocris Bioscience | 1045 |
| D-AP5 | Tocris Bioscience | 0106 |
| **Software** | | |
| PATCHMASTER | HEKA Elektronik | N/A |
| pClamp | Molecular Devices | 10.0 |
| Igor | Wavemetrics | 5.03 |
| MATLAB | mathworks | 2017b |
| **Other** | | |
| Vibrating microtome | Leica | VT 1000S |

| Reagent/resource | Reference or source | Identifier or catalog number |
|---|---|---|
| EPC amplifier | HEKA | 10/2 |
| Photometry | ThinkerTech Nanjing Biotech Ltd | N/A |
| Confocal microscope | Zeiss | LSM 880 |

## Animals

All of the experimental procedures were reviewed and approved by the Animal Advisory Committee of Northeast Normal University (NENU/IACUC, AP20151009). All of the mice were housed under a 12/12 h light–dark cycle, with lights on from 6:00 to 18:00 each day. They were provided with food and water ad libitum.

Adult mice aged between 12 weeks and 20 weeks were used in all of our studies. C57BL/6J mice were purchased from the Yisi Animal Center located in Changchun, China. The Tac1-IRES2-Cre mouse strain (Jax No. 021877) was purchased from the Jackson Laboratory.

## Viruses

For retrograde circuit tracing, AAV-EF1α-DIO-EGFP-T2A-TVA (AAV2/9, $3.00 \times 10^{12}$ particles ml$^{-1}$), AAV-EF1α-DIO-oRVG (AAV2/9, $5.33 \times 10^{12}$ particles ml$^{-1}$) and RV-ENVA-ΔG-mCherry ($2.00 \times 10^{8}$ particles ml$^{-1}$) were purchased from Brain Case (China).

For fiber photometry, AVV-CAG-Retro-Cre (AAV2/9, $3.20 \times 10^{12}$ particles ml$^{-1}$) and AAV-EF1α-DIO-GCaMP6m (AAV2/9, $2.22 \times 10^{12}$ particles ml$^{-1}$) were purchased from Brain Case (China).

For functional analysis, AAV-EF1α-DIO-hM4D(Gi)-mCherry (AAV2/9, $6.044 \times 10^{12}$ particles ml$^{-1}$), AAV-EF1α-DIO-hM3D(Gq)-mCherry (AAV2/9, $7.205 \times 10^{12}$ particles ml$^{-1}$), AAV-EF1α-DIO-mCherry (AAV2/9, $5.47 \times 10^{12}$ particles ml$^{-1}$), AAV-EF1α-fDIO-hM4D(Gi)-mCherry (AAV9, $5.47 \times 10^{12}$ particles ml$^{-1}$), AAV-EF1α-fDIO-hM3D(Gq)-mCherry (AAV9, $5.41 \times 10^{12}$ particles ml$^{-1}$), AAV-EF1α-fDIO-mCherry (AAV9, $4.14 \times 10^{12}$ particles ml$^{-1}$), AAV-Retro-CAG-FLEX-flpo (Retro, $5.0 \times 10^{12}$ particles ml$^{-1}$), AAV-EF1α-DIO-flpo (AAV1, $2.0 \times 10^{13}$ particles ml$^{-1}$), AVV-CAG-Retro-Cre (AAV2/9, $3.20 \times 10^{12}$ particles ml$^{-1}$) and AAV-EF1α-DIO-ChR2-mCherry (AAV9, $3.20 \times 10^{12}$ particles ml$^{-1}$) were purchased from Brain Case (China).

For Tac1 knockdown, the following short hairpin sequence was used: 5'-GTTCTTTGGATTAATGGGCAA-3'; the control scramble sequence was 5'-CGCTGAGTACTTCGAAATGTC-3'. AAV-scramble-shRNA-mCherry (AAV2/9, $3.6 \times 10^{12}$ particles ml$^{-1}$) and AAV-TAC1-shRNA-mCherry (AAV2/9, viral: $4.01 \times 10^{12}$ particles ml$^{-1}$) were validated and produced by Brain Case (China).

## Stereotaxic surgery

The animals were first anesthetized via 1.0% sodium pentobarbital (0.1 g/kg body weight, i.p.). Viruses were injected into specific regions of the animals' brains at a rate of 50 nl/min, which was achieved via a stereotaxic instrument (RWD Co., China) and a 5-μl syringe (Hamilton, Sigma). After the injection was complete, the syringe was held in place for an additional 10 min before being slowly withdrawn. All of the behavioral experiments were performed 4 weeks after the injection. The coordinates for the injections were determined via the Paxinos and Franklin Mouse Brain Atlas and were empirically adjusted. The coordinates for injection into the MeA were −1.35 mm AP, ±2.00 mm ML, and −5.25 mm DV. The coordinates for injection into the VMHvl were −1.46 mm AP, ±0.6 mm ML, and −5.6 mm DV. The coordinates for injection into the VTA were −3.20 mm AP, ±0.25 mm ML, and −4.4 mm DV.

For fiber photometric recording, 150 nl of AAV-EF1α-DIO-GCaMP6m was unilaterally injected into the MeA. For chemogenetic manipulation, 200 nl of AAV-EF1α-DIO-hM4D(Gi)-mCherry, AAV-EF1α-DIO-hM3D(Gq)-mCherry or AAV-EF1α-DIO-mCherry were bilaterally injected into the MeA; 200 nl of AAV-EF1α-fDIO-hM4D(Gi)-mCherry, AAV-EF1α-DIO-fhM3D(Gq)-mCherry or AAV-EF1α-fDIO-mCherry were bilaterally injected into the MeA; 200 nl of AAV-Retro-CAG-FLEX-flpo were bilaterally injected into the VMHvl; 200 nl of AAV-EF1α-fDIO-hM4D(Gi)-mCherry, AAV-EF1α-fDIO-hM3D(Gq)-mCherry or AAV-EF1α-fDIO-mCherry were bilaterally injected into the MeA; and 200 nl of AAV1-EF1α-DIO-flpo were bilaterally injected into the VTA. 200 nl of AAV-EF1α-DIO-hM4D(Gi)-mCherry, AAV-EF1α-DIO-hM3D(Gq)-mCherry or AAV-EF1α-DIO-mCherry were bilaterally injected into the VTA; and 200 nl of AAV1-Retro-Cre were bilaterally injected into the MeA. output mapping, 150 nl of AAV-EF1α-DIO-mCherry was unilaterally injected into the MeA. For retrograde circuit mapping of MeA$^{Tac1}$ neurons, 50 nl of AAV-EF1α-DIO-EGFP-T2A-TVA and 250 nl of AAV-EF1α-DIO-oRVG were unilaterally injected into the MeA. Three weeks later, 250 nl of RV-ENVA-ΔG-mCherry was injected into the same coordinates. For electrophysiological recording, 200 nl of AAV-EF1α-DIO-ChR2-mCherry was unilaterally injected into the MeA.

## Fiber photometry

A fiberoptic cannula (ferrule: Φ2.5; fiber: 200-μm OD, 0.37 NA, 5.5 mm long) was implanted into the MeA. The cannula was securely attached to the skull using dental acrylic, and a skull-penetrating M1 screw was used for stability. For the recordings, we utilized a photometry system provided by ThinkerTech Nanjing Biotech Ltd.

To prevent bleaching of GCaMP, we calibrated the laser power at the fiber tip to a relatively low level (405 nm, 20 μW; 470 nm, 40 μW). Data from the fiber photometry were subsequently analyzed via a dedicated program provided by ThinkerTech, which is based on the MATLAB platform. This involved the alignment of the timing of the behavior with the GCaMP signal and the extraction of a data epoch from −10 s to +10 s around the experiments.

The data processing included correcting the raw fluorescence data (F) with the airPLS algorithm (lambda = 8), followed by segmenting and aligning the data with the onset of behavioral events within individual trials or sessions. We calculated the relative fluorescence change values (ΔF/F) as (F-F0)/(F0-Voffset), where F0 represents the baseline fluorescence signal (averaged over the −2 s to 0 s time window prior to a trigger event), and Voffset is the fluorescence signal that was recorded prior to connecting the cannula to the optical fiber above the MeA. The resulting ΔF/F values were visualized as heatmaps or as average plots, with shaded areas indicating the standard error of the mean (SEM). During the elevated plus maze test, we estimated the instantaneous ΔF/F

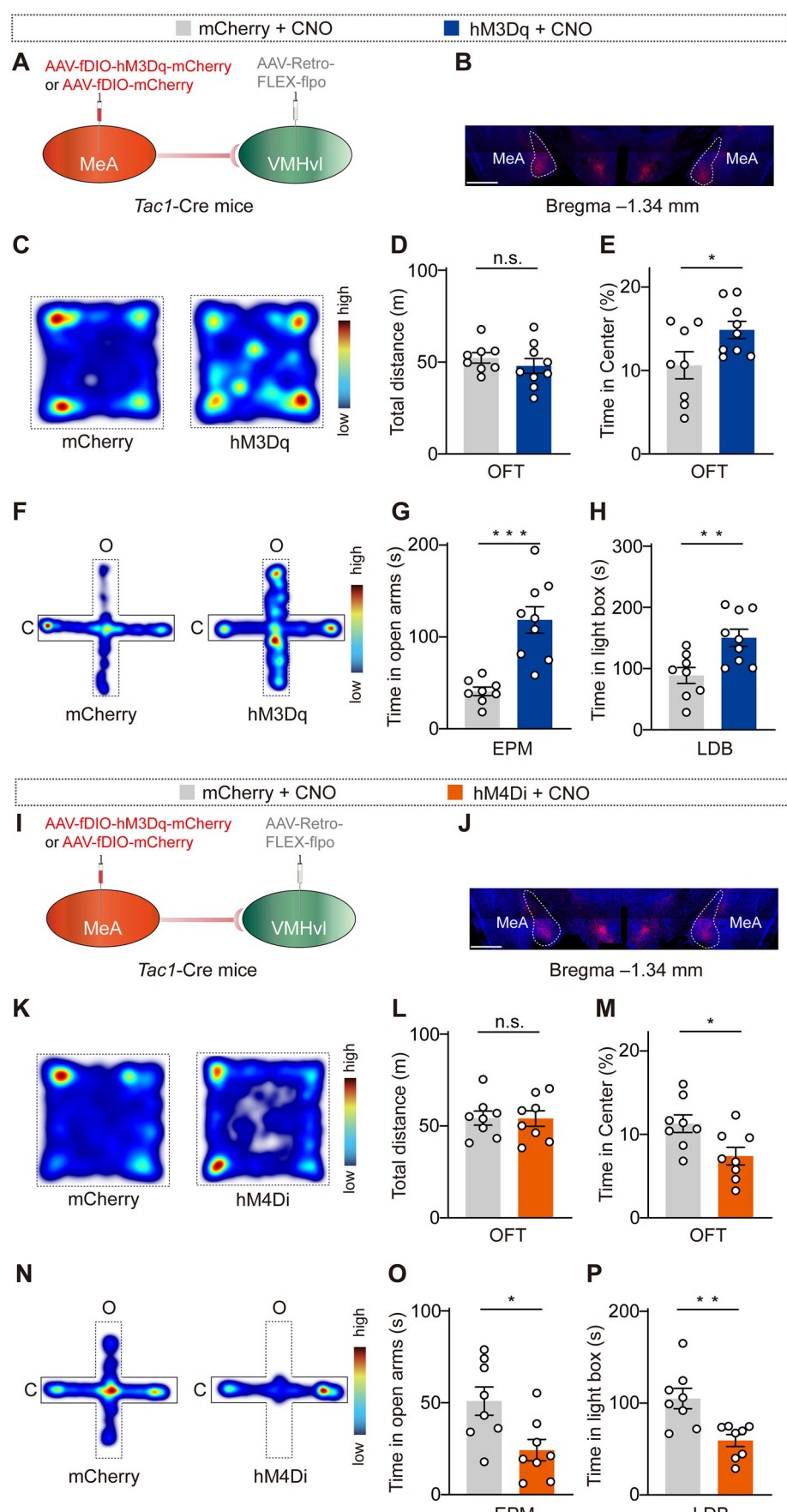

◄  **Figure 5.   The MeA^Tac1→VMHvl pathway regulates anxiety-like behaviors in male mice.**

(A) Schematic of the strategies that were used to express AAV-Retro-FLEX-flpo in the VMHvl and AAV-fDIO-mCherry or AAV-fDIO-hM3Dq-mCherry in the MeA of *Tac1*-Cre mice. (B) Representative images showing virus expression in the MeA. Scale bar, 1 mm. (C) Heatmaps displaying the time spent in different regions of the open field arena (warmer colors indicate more time spent in a region). (D) Total distance traveled in the open field arena. *P* = 0.3987. mCherry: *N* = 8; hM3Dq: *N* = 9. (E) Activation of Tac1 neurons increased the time spent in the center of the arena. *P* = 0.0397. mCherry: *N* = 8; hM3Dq: *N* = 9. (F) Heatmaps displaying the time spent in different regions of the elevated plus maze (warmer colors indicate more time spent in a region). (G) Activation of Tac1 neurons increased the time spent in the open arms. *P* = 0.0002. mCherry: *N* = 8; hM3Dq: *N* = 9. (H) Time spent by mice with activated Tac1 neurons in the light box. *P* = 0.0061. mCherry: *N* = 8; hM3Dq: *N* = 9. (I) Schematic of the strategies that were used to express AAV-Retro-FLEX-flpo in the VMHvl and AAV-fDIO-mCherry or AAV-fDIO-hM4Di-mCherry in the MeA of *Tac1*-Cre mice. Scale bar, 1 mm. (J) Representative images showing virus expression in the MeA. Scale bar, 1 mm. (K) Heatmaps displaying the time spent in different regions of the open field arena (warmer colors indicate more time spent in a region). (L) Total distance traveled in the open field arena. *P* = 0.9677. mCherry: *N* = 8; hM4Di: *N* = 8. (M) Inhibition of Tac1 neurons reduced the time spent in the center of the arena. *P* = 0.0208. mCherry: *N* = 8; hM4Di: *N* = 8. (N) Heatmaps displaying the time spent in different regions of the elevated plus maze (warmer colors indicate more time spent in a region). (O) Inhibition of Tac1 neurons reduced the time spent in the open arms. *P* = 0.0150. mCherry: *N* = 8; hM4Di: *N* = 8. (P) Inhibition of Tac1 neurons reduced the time spent in the light box. *P* = 0.0031. mCherry: *N* = 8; hM4Di: *N* = 8. *N* animal number, MeA medial amygdala. Statistical significance was determined via an unpaired *t* test. All of the data are presented as the means ± s.e.m.s. *P < 0.05; **P < 0.01; ***P < 0.001; n.s. not significant. See Table EV1 for detailed statistics. Source data are available online for this figure.

during transitions between the open arms and closed arms of the maze.

While the mice explore the elevated plus maze, the EV recording software will simultaneously record the mouse movement trajectories captured by EthoVision XT 11 and the calcium signal changes recorded by the dual-channel software, ensuring that the occurrence of mouse events corresponds with the fluctuations in calcium signals. The time points when the mice just enter the open arm or the closed arm will be marked. All events are marked as starting from 0 s, and calcium signal analysis is conducted within a 10-s window before and after the event. The averaged calcium signal data for each mouse are directly calculated by MATLAB 2017b as the Peak and AUC.

## In vivo chemogenetic experiments

For chemogenetic manipulations, the mice were given an intraperitoneal injection of clozapine N-oxide dihydrochloride (CNO) at a dose of 5 mg/kg. After 30 min, the mice were subjected to behavioral experiments.

## Immunohistochemistry

The mice were anesthetized via sodium pentobarbital and transcardially perfused with 0.1 M phosphate-buffered saline (PBS), followed by 4% paraformaldehyde (PFA) solution in PBS. The brains were subsequently postfixed at 4 °C overnight and then immersed in a 30% sucrose solution for preservation.

The brain sections were meticulously sliced at a thickness of 40 μm via a freezing microtome (Leica, CM 1950). The sections were rinsed with PBS and then treated with a blocking solution consisting of 0.2% Triton X-100, 10% serum, and 2% bovine serum albumin (BSA) in 0.1 M PBS for 2 h. Following thorough washing with PBS, the sections were counterstained with DAPI (1:2000, Life Technologies, D3571) for 8 min to highlight the nuclei. The sections were subsequently preserved in ProLong Gold mounting media (Thermo Fisher, P36930).

The following primary antibodies were used: Rabbit anti-VLGUT2 (1:200; 135 402, Synaptic Systems, Germany) Rat anti-GAD67 (1:500; ab75712, Abcam, USA) and Rabbit anti-Tyrosine Hydroxylase (1:1000; ab6211, Abcam, USA). The following secondary antibodies were used: Alexa Fluor 488-conjugated

goat anti-rabbit (1:1000; A11008, Invitrogen, USA). All images were acquired with a Zeiss LSM 880 confocal microscope (Germany).

## Behavioral tests

All of the mice that were used for the behavioral assays were male and their littermates, unless otherwise mentioned. An experimenter who was blinded to the genotypes performed all of the tests.

### Open field test
The mice were introduced from one corner of the experimental area (40 cm × 40 cm × 40 cm). During the experiment, the mice were allowed to freely explore the field for 10 min. The EthoVision XT system was utilized to record the movements of the mice, from which the total distance traveled and exploration time in the central area were statistically analyzed.

### Elevated plus maze
Mice were introduced into the center of the plus maze apparatus containing two closed arms ($25 \times 5 \times 20\ cm^3$) facing each other, as well as two open arms ($25 \times 5\ cm^2$). The apparatus was 80 cm high. Mice were allowed to freely explore the maze for 5 min. The time spent in each arm was scored via the EthoVision XT system.

### Light/dark box
Mice were positioned in the light chamber of the apparatus ($15 \times 20 \times 20\ cm^3$). Mice were allowed to explore light and dark chambers freely for 5 min. The duration spent in each chamber was measured via the EthoVision XT system.

### Tail suspension test
Mice were suspended by their tails at 20 cm above the ground and recorded for 5 min via the EthoVision XT system. The time spent immobile was scored, and animals displaying climbing behavior were excluded from the analysis.

### Sucrose preference test
Mice were first given 96 h of adaptation and baseline measurement with two identical bottles. One bottle contained water, and the other bottle contained a 1.5% sucrose solution. Subsequently, the mice were deprived of water for 24 h. After deprivation, mice were

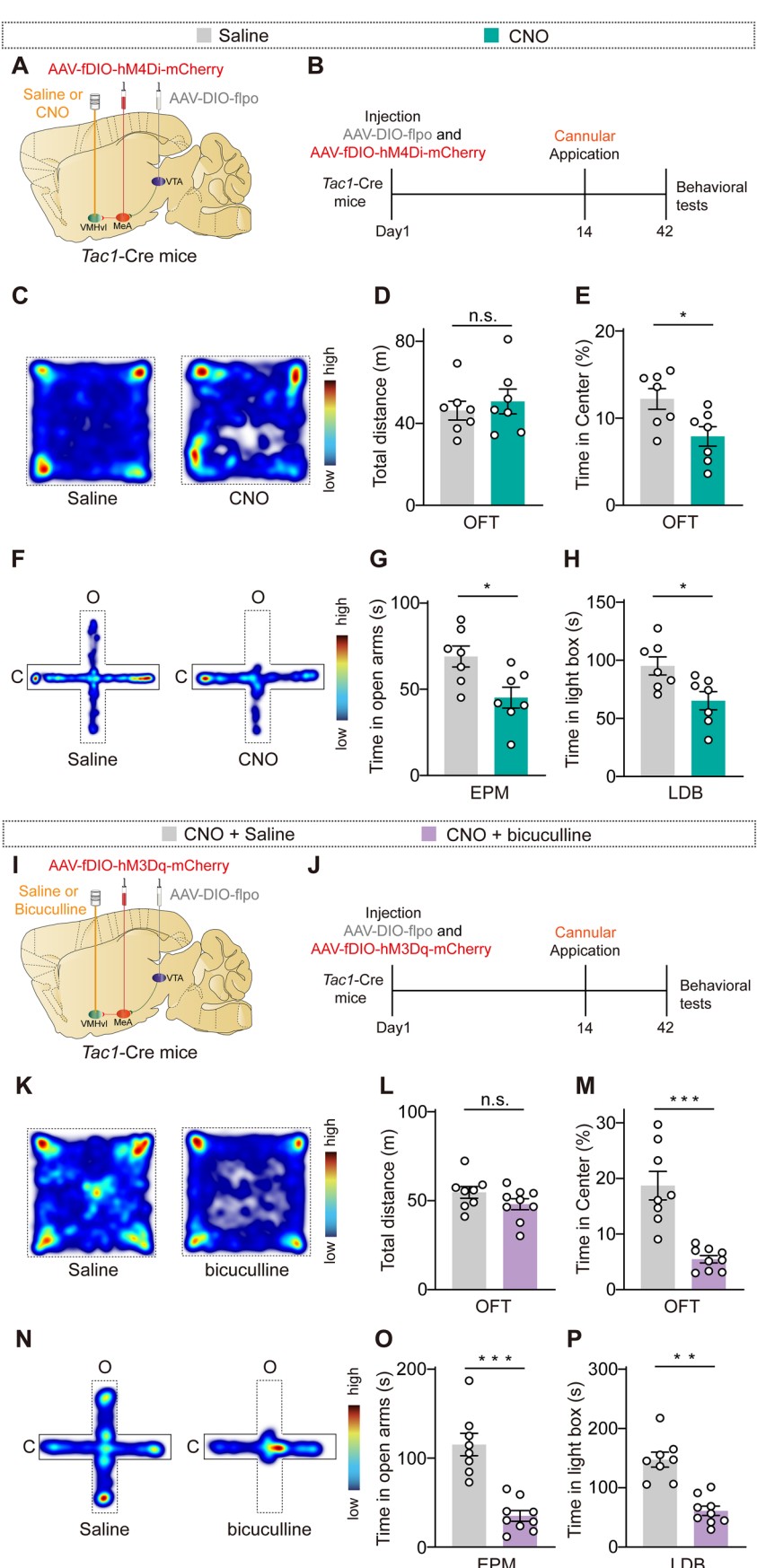

**Figure 6. The VTA→MeA^Tac1→VMHvl circuit regulates anxiety-like behaviors in male mice.**

(A) Schematic of the experimental strategies. (B) Experimental timeline. (C) Heatmaps displaying the time spent in different regions of the open field arena (warmer colors indicate more time spent in a region). (D) Total distance traveled in the open field arena. $P = 0.5679$. Saline: $N = 7$; CNO: $N = 7$. (E) Time spent in the center of the arena. $P = 0.0216$. Saline: $N = 7$; CNO: $N = 7$. (F) Heatmaps displaying the time spent in different regions of the elevated plus maze (warmer colors indicate more time spent in a region). (G) Time spent in the open arms. $P = 0.0163$. Saline: $N = 7$; CNO: $N = 7$. (H) Time spent in the light box. $P = 0.0187$. Saline: $N = 7$; CNO: $N = 7$. (I) Schematic of the experimental strategies. (J) Experimental timeline. (K) Heatmaps displaying the time spent in different regions of the open field arena (warmer colors indicate more time spent in a region). (L) Total distance traveled in the open field arena. $P = 0.1633$. Saline: $N = 8$; bicuculline: $N = 9$. (M) Time spent in the center of the arena. $P < 0.0001$. Saline: $N = 8$; bicuculline: $N = 9$. (N) Heatmaps displaying the time spent in different regions of the elevated plus maze (warmer colors indicate more time spent in a region). (O) Time spent in the open arms. $P < 0.0001$. Saline: $N = 8$; bicuculline: $N = 9$. (P) Time spent in the light box. $P < 0.0001$. Saline: $N = 8$; bicuculline: $N = 9$. N animal number, MeA medial amygdala. Statistical significance was determined via an unpaired $t$ test. All of the data are presented as the means ± s.e.m.s. $**P < 0.01$; $***P < 0.001$; n.s. not significant. See Table EV1 for detailed statistics. Source data are available online for this figure.

presented with sucrose and water. The amount of consumed liquid (sucrose or water) was measured at 24 h.

## Ex vivo electrophysiology

Mice were anesthetized with sodium pentobarbital and quickly decapitated to remove the brain. Acute slices (300 μm thick) were cut by using a vibrating microtome (Leica, VT 1000S). The sections were quickly transferred to a recovery chamber and incubated at 35 °C for 30 min in recovery solution comprising 93 mM NMDG, 1.2 mM $NaH_2PO_4$, 30 mM $NaHCO_3$, 20 mM HEPES, 25 mM D-glucose, 5 mM Na-ascorbate, 2 mM thiourea, 3 mM Na-pyruvate, 3 mM KCl, 10 mM $MgSO_4$, 0.5 mM $CaCl_2$, 93 mM HCl and 12 mM NAC (pH 7.4). The slices were then incubated at room temperature for 1 h in carbogenated artificial cerebrospinal fluid (aCSF) comprising 120 mM NaCl, 2.5 mM KCl, 1.0 mM $NaH_2PO_4$, 26 mM $NaHCO_3$, 11 mM D-glucose, 2.0 mM $MgCl_2$, and 2.0 mM $CaCl_2$ (pH 7.4) before recording. All of the solutions were saturated with 95% $O_2$/5% $CO_2$.

Whole-cell patch-clamp recordings were performed via an EPC-10/2 amplifier (HEKA, Germany). The recording pipettes were pulled from borosilicate glass tubes (Sutter Instruments) and had a resistance of 3–6 MΩ; only whole-cell patches with a series resistance <15 MΩ were used for the recordings. The IPSCs were recorded by holding the membrane potential at −70 mV.

For optical recording, AAV-DIO-ChR2-mCherry was injected into the MeA of Tac1-Cre mice, and VMHvl neurons in areas with a high density of mCherry terminals were patched. ChR2 with 465-nm blue light was delivered via a laser (INPER-B1–465, INPER, China). To record optically evoked IPSCs (oIPSCs) in VMHvl neurons, CNQX (50 μM, Tocris Bioscience, 1045) was added to the aCSF. Patch pipettes were filled with 135 mM CsCl, 1 mM EGTA, 4 mM Mg-ATP, 0.6 mM Na-GTP and 10 mM HEPES (pH 7.4).

Data were acquired via PATCHMASTER software (HEKA, Germany), Clampfit (Molecular Devices), and Igor (Wavemetrics) software.

## Quantification and statistical analyses

All of the experimental procedures and data analyses were conducted in a blinded manner. The number of replicates ($N$ or $n$) indicated in the figure legends refers to the number of experimental subjects that were independently treated in each experiment. All of the statistical analyses were performed in GraphPad Prism (GraphPad Software), unless otherwise mentioned. Student's $t$ tests and two-way analyses of variance (ANOVAs) were used to conduct group comparisons. Statistical significance was set at $*P < 0.05$, $**P < 0.01$, and

$***P < 0.001$. The data are presented as the means ± standard error of the means (s.e.m.s).

## Data availability

This study includes no data deposited in external repositories. The data that support the findings of this study are available from the corresponding author.

The source data of this paper are collected in the following database record: biostudies:S-SCDT-10_1038-S44319-025-00528-z.

## Peer review information

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

## Acknowledgements

This work was supported by the STI 2030—Major Projects 2021ZD0204000 (2021ZD0204003), the National Natural Science Foundation of China (32071018) and the Jilin Provincial Department of Education (JJKH20231313KJ).

## Author contributions

**Yao Wang**: Data curation; Investigation; Methodology. **Jiu-Ye Qiao**: Conceptualization; Data curation; Investigation; Methodology. **Mei-Hui Yue**: Data curation; Software; Formal analysis; Validation. **Xin-Yue Lv**: Project administration. **Si-Ran Wang**: Validation; Visualization. **Qian-Qian Yang**: Software. **Han-Yun Kang**: Methodology. **Hua-Li Yu**: Software. **Xiao-Xiao He**: Software. **Xiao-Juan Zhu**: Supervision; Funding acquisition; Writing—original draft. **Zi-Xuan He**: Supervision; Funding acquisition; Investigation; Writing—original draft; Writing—review and editing.

Source data underlying figure panels in this paper may have individual authorship assigned. Where available, figure panel/source data authorship is listed in the following database record: biostudies:S-SCDT-10_1038-S44319-025-00528-z.

## Disclosure and competing interests statement

The authors declare no competing interests.

# Expanded View Figures

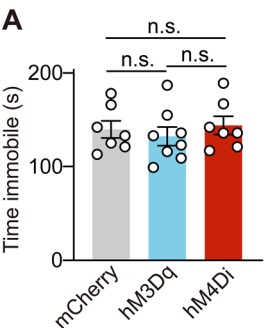 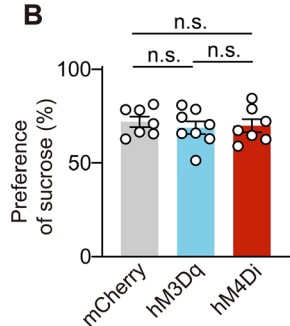

**Figure EV1. MeATac1 neurons do not modulate depressive-like behaviors in male mice.**

(A) The immobile time in the tail suspension test. (B) The preference score in the sucrose preference test. mCherry: $N = 7$; M3Dq: $N = 8$; hM4Di: $N = 7$. $N$ animal number. All data are means ± s.e.m. n.s. not significant. See Table EV1 for detailed statistics. Source data are available online for this figure.

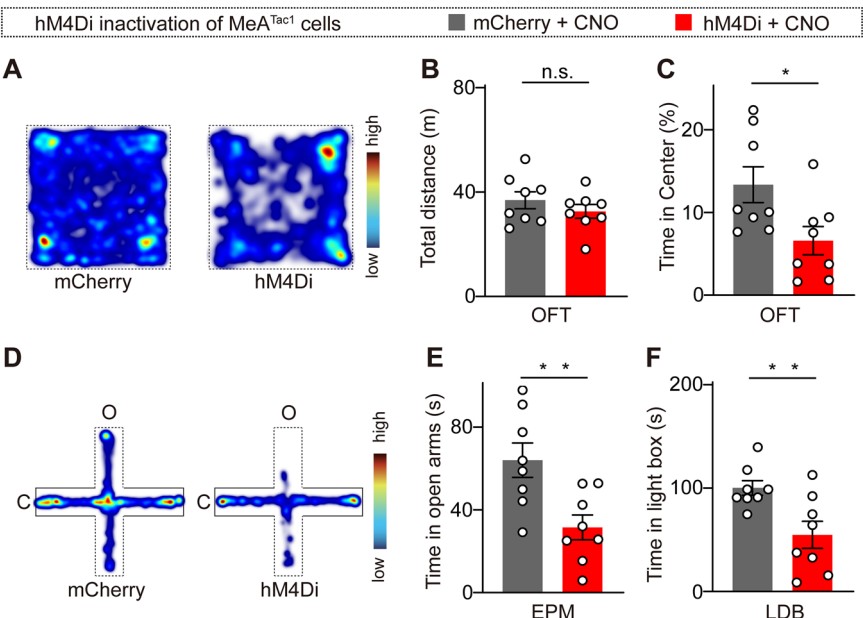

**Figure EV2. MeATac1 neurons mediate anxiety-like behaviors in female mice.**

(A) Heatmaps display time spent in different regions of the open field arena (warmer colors indicate more time). (B) Total distance traveled in the open field arena. $P = 0.3245$. mCherry: $N = 8$; hM4Di: $N = 8$. (C) Decreased time spent in the center of the open field arena was observed in hM4Di female mice. $P = 0.0275$. mCherry: $N = 8$; hM4Di: $N = 8$. (D) Heatmaps display time spent in different regions of the elevated plus maze (warmer colors indicate more time). (E) Decreased time spent in the open arms was observed in hM4Di female mice. $P = 0.0068$. mCherry: $N = 8$; hM4Di: $N = 8$. (F) Decreased time spent in the light box was observed in hM4Di female mice. $P = 0.0088$. mCherry: $N = 8$; hM4Di: $N = 8$. N animal number. Statistical significance was determined using an unpaired $t$ test. All data are means ± s.e.m. *$P < 0.05$; **$P < 0.01$; n.s. not significant. See Table EV1 for detailed statistics. Source data are available online for this figure.

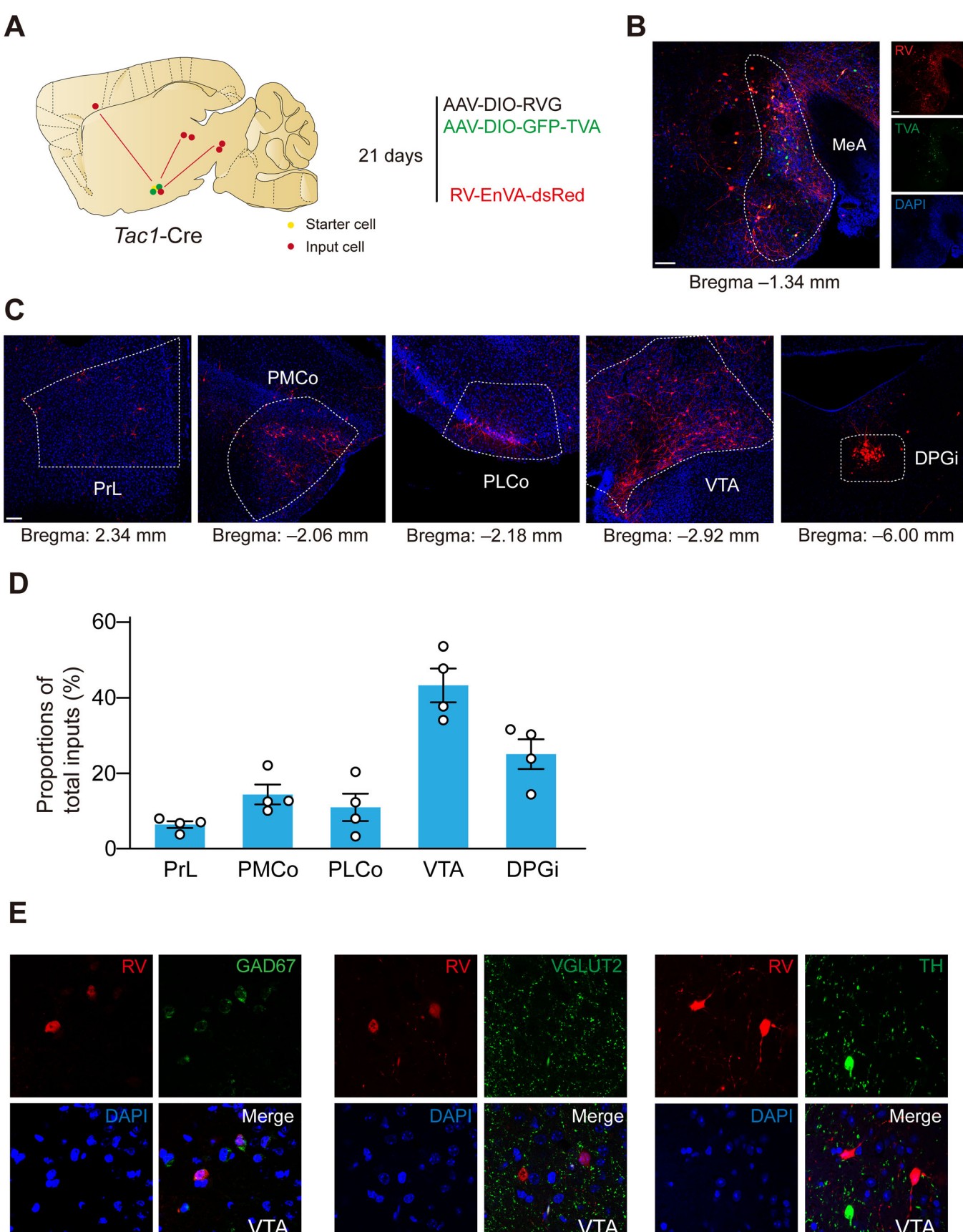

◀ **Figure EV3. Whole-brain inputs to Tac1 neurons in the MeA.**

(A) Schematic of rabies-based cell type-specific monosynaptic tracing procedure. (B) Starter neurons with AAV-EF1α-DIO-EGFP-T2A-TVA, AAV-EF1α-DIO-oRVG and RV-ENVA-ΔG-mCherry. Scale bar, 100 μm. (C) Representative images of tracing inputs from selected brain regions to MeATac1 neurons. PrL prelimbic cortex, PMCo posteromedial cortical amygdaloid area, PLCo posterlateral cortical amygdaloid area, VTA ventral tegmental area, DPGi dorsal paragigantocellular nucleus. Scale bar, 100 μm. (D) Whole brain mapping quantitation of inputs to MeATac1 neurons. $N = 4$. N animal number. (E) Representative images of acute slices injected with RV-ENVA-ΔG-mCherry staining with antibodies against GAD67, VGLUT2 and TH. Scale bar, 10 μm. Error bars represent s.e.m. Source data are available online for this figure.

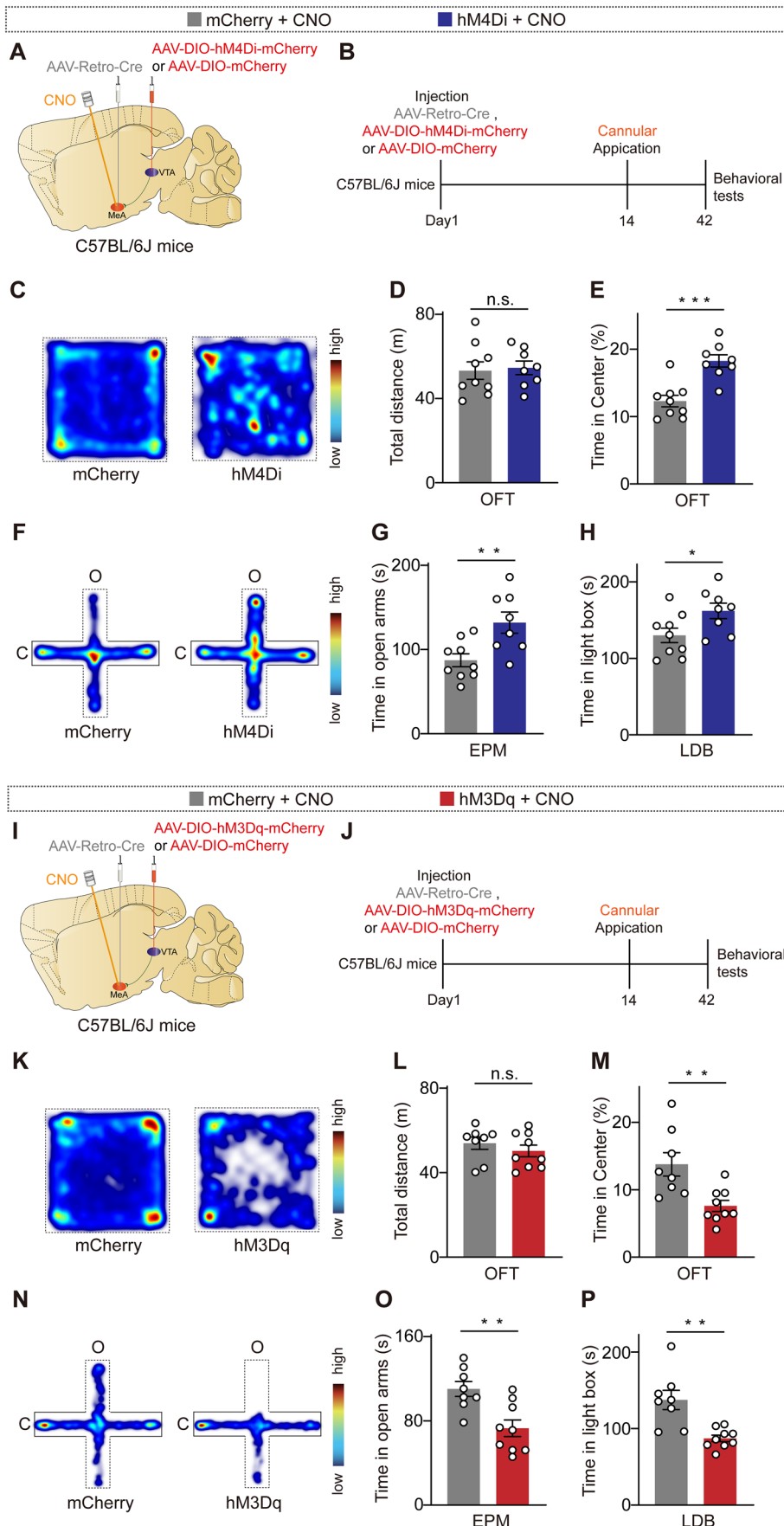

◄ **Figure EV4. The VTA→MeATac1 pathway modulates anxiety-like behaviors.**

(A, B) Schematic of the strategies that were used to express AAV-Retro-Cre in the MeA and AAV-DIO-mCherry or AAV-DIO-hM4Di-mCherry in the VTA of C57BL/6 J mice. A cannula was then implanted into the MeA. (C) Heatmaps displaying the time spent in different regions of the open field arena (warmer colors indicate more time spent in a region). (D) Total distance traveled in the open field arena. $P = 0.8089$. mCherry: $N = 9$; hM3Dq: $N = 8$. (E) Time spent in the center of the arena. $P = 0.003$. mCherry: $N = 9$; hM3Dq: $N = 8$. (F) Heatmaps displaying the time spent in different regions of the elevated plus maze (warmer colors indicate more time spent in a region). (G) Time spent in the open arms. $P = 0.0068$. mCherry: $N = 9$; hM3Dq: $N = 8$. (H) Time spent in the light box. $P = 0.0345$. mCherry: $N = 9$; hM3Dq: $N = 8$. (I, J) Schematic of the strategies that were used to express AAV-Retro-Cre in the MeA and AAV-DIO-mCherry or AAV-DIO-hM3Dq-mCherry in the VTA of C57BL/6 J mice. A cannula was then implanted into the MeA. (K) Heatmaps displaying the time spent in different regions of the open field arena (warmer colors indicate more time spent in a region). (L) Total distance traveled in the open field arena. $P = 0.3819$. mCherry: $N = 8$; hM4Di: $N = 9$. (M) Time spent in the center of the arena. $P = 0.0047$. mCherry: $N = 8$; hM4Di: $N = 9$. (N) Heatmaps displaying the time spent in different regions of the elevated plus maze (warmer colors indicate more time spent in a region). (O) Time spent in the open arms. $P = 0.0032$. mCherry: $N = 8$; hM4Di: $N = 9$. (P) Time spent in the light box. $P = 0.0012$. mCherry: $N = 8$; hM4Di: $N = 9$. $N$ animal number. Statistical significance was determined via an unpaired $t$ test. All of the data are presented as the means ± s.e.m.s. *$P < 0.05$; **$P < 0.01$; ***$P < 0.001$; n.s., not significant. See Table EV1 for detailed statistics. Source data are available online for this figure.

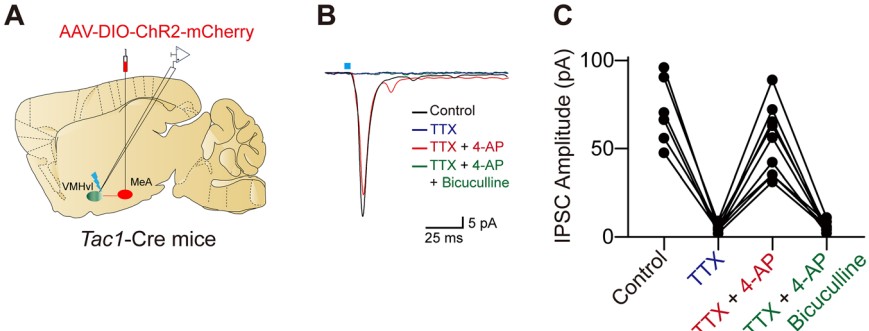

**Figure EV5. Identifying the neurotransmission of MeATac1→VMHvl pathway.**

(A) Schematic of optogenetic electrophysiological recording procedure. (B, C) Representative traces and quantification of the amplitudes. $n = 8$ cells from 4 animals. MeA medial amygdala. Source data are available online for this figure.

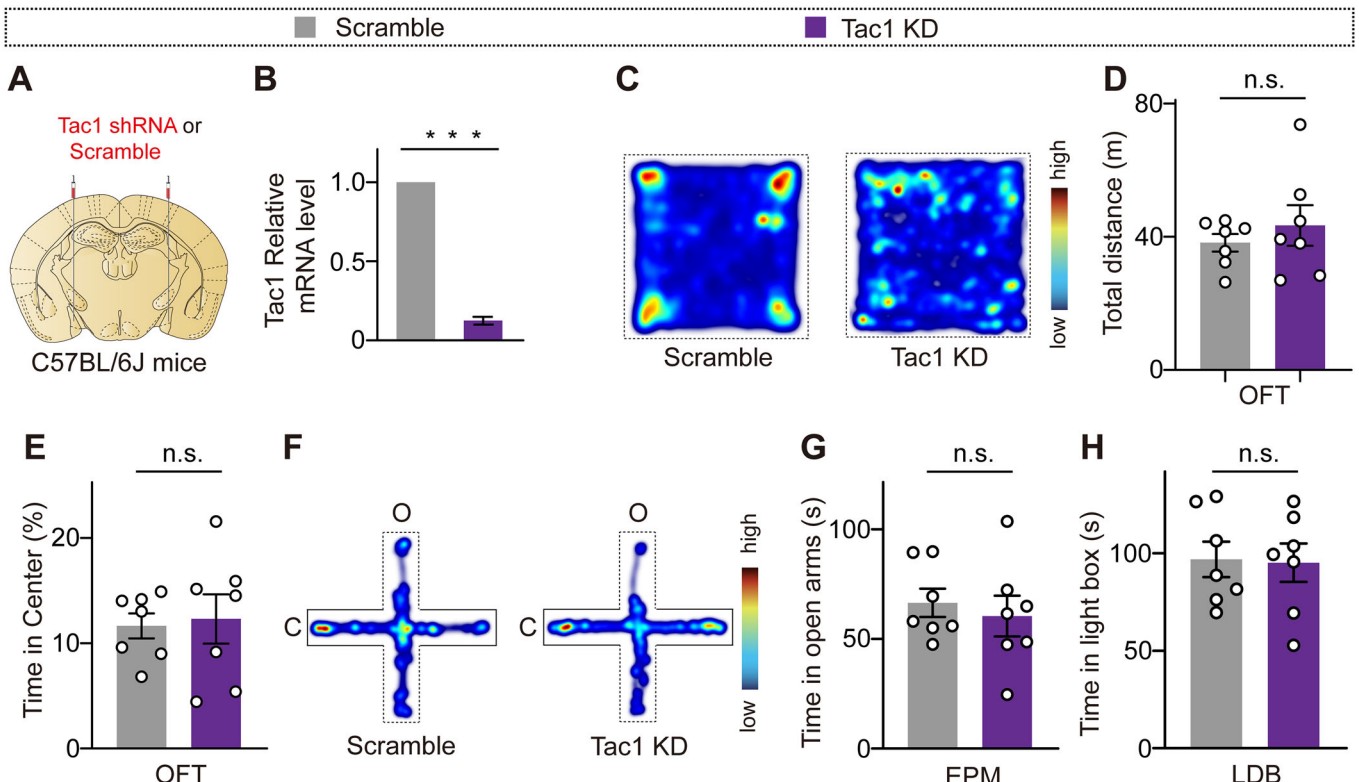

**Figure EV6. Substance P does not affect anxiety-like behaviors in mice.**

(A) Schematic of strategies used to express AAV-Tac1-shRNA and scramble shRNA in C57BL/6J mice. (B) Quantitative RT–PCR confirmed the reduction in Tac1 after knockdown. (C) Heatmaps display time spent in different regions of the open field arena (warmer colors indicate more time). (D) Total distance traveled in the open field arena. $P = 0.4478$. Scramble: $N = 7$; Knockdown: $N = 7$. (E) Time spent in the center of the open field arena. $P = 0.8068$. Scramble: $N = 7$; Knockdown: $N = 7$. (F) Heatmaps display time spent in different regions of the elevated plus maze (warmer colors indicate more time). (G) Time spent in the open arms. $P = 0.6055$. Scramble: $N = 7$; Knockdown: $N = 7$. (H) Time spent in the light box. $P = 0.8986$. Scramble: $N = 7$; Knockdown: $N = 7$. Statistical significance was determined using an unpaired $t$ test. All data are means ± s.e.m. n.s., not significant. See Table EV1 for detailed statistics. Source data are available online for this figure.

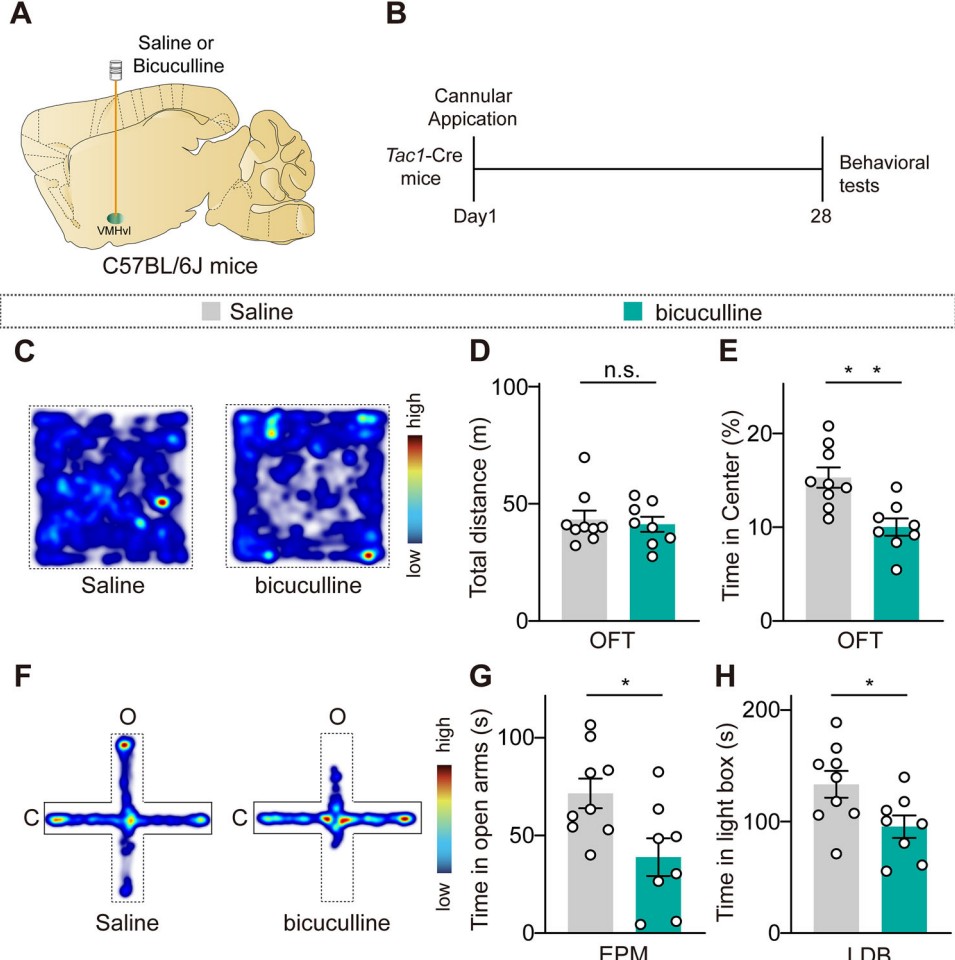

**Figure EV7. GABA regulates anxiety-like behaviors in mice.**

(A) Schematic of experimental strategies. (B) Experimental timeline. (C) Heatmaps display time spent in different regions of the open field arena (warmer colors indicate more time). (D) Total distance traveled in the open field arena. $P = 0.6904$. saline: $N = 9$; bicuculline: $N = 8$. (E) Time spent in the center of the arena. $P = 0.0024$. saline: $N = 9$; bicuculline: $N = 8$. (F) Heatmaps display time spent in different regions of the elevated plus maze (warmer colors indicate more time). (G) Time spent in the open arms. $P = 0.0171$. saline: $N = 9$; bicuculline: $N = 8$. (H) Time spent in the light box. $P = 0.0308$. saline: $N = 9$; bicuculline: $N = 8$. $N$ animal number. Statistical significance was determined using an unpaired $t$ test. All data are means ± s.e.m. *$P < 0.05$; **$P < 0.01$; n.s., not significant. See Table EV1 for detailed statistics. Source data are available online for this figure.

