## [Peer Review File · EMBO Reports]

A MeA Tac1 Neural Circuit Mediates Anxiety-like Behaviors in Mice

Zi-Xuan He, Yao Wang, Jiu-Ye Qiao, Mei-Hui Yue, Xin-Yue Lv, Si-Ran Wang, Qian-Qian Yang, Han-Yu Kang, Huali Yu, Xiaoxiao He, and Xiaojuan Zhu

Corresponding author(s): Zi-Xuan He (hez234@nenu.edu.cn)

Review Timeline:

Submission Date:	2nd Dec 24
Editorial Decision:	21st Jan 25
Revision Received:	16th Apr 25
Editorial Decision:	27th May 25
Revision Received:	8th Jun 25
Accepted:	4th Jul 25

Editor: Esther Schnapp

Transaction Report:

Dear Dr. He,

Thank you for the submission of your manuscript to EMBO reports. We have now received the full set of referee reports that is pasted below.

As you will see, the referees acknowledge that the findings are potentially interesting. However, they also have several suggestions for how the study should be improved and I think that all points should be addressed. Please let me know in case you disagree and we can discuss the exact revision requirements further, also in a video chat, if you like.

I would thus like to invite you to revise your manuscript with the understanding that the referee concerns must be fully addressed and their suggestions taken on board. Please address all referee concerns in a complete point-by-point response. Acceptance of the manuscript will depend on a positive outcome of a second round of review. It is EMBO reports policy to allow a single round of major revision only and acceptance or rejection of the manuscript will therefore depend on the completeness of your responses included in the next, final version of the manuscript.

We realize that it is difficult to revise to a specific deadline. In the interest of protecting the conceptual advance provided by the work, we recommend a revision within 3 months (23rd Apr 2025). Please discuss the revision progress ahead of this time with the editor if you require more time to complete the revisions.

- 1) A data availability section providing access to data deposited in public databases is missing. If you have not deposited any data, please add a sentence to the data availability section that explains that.
- 2) Your manuscript contains statistics and error bars based on $n=2$. Please use scatter blots in these cases. No statistics should be calculated if $n=2$.

5) a complete author checklist, which you can download from our author guidelines <https://www.embopress.org/page/journal/14693178/authorguide>. Please insert information in the checklist that is also reflected in the manuscript. The completed author checklist will also be part of the RPF.

6) Please note that all corresponding authors are required to supply an ORCID ID for their name upon submission of a revised manuscript (<https://orcid.org/>). Please find instructions on how to link your ORCID ID to your account in our manuscript tracking system in our Author guidelines <https://www.embopress.org/page/journal/14693178/authorguide#authorshipguidelines>

7) Before submitting your revision, primary datasets produced in this study need to be deposited in an appropriate public

database (see <https://www.embopress.org/page/journal/14693178/authorguide#datadeposition>). Please remember to provide a reviewer password if the datasets are not yet public. The accession numbers and database should be listed in a formal "Data Availability" section placed after Materials & Method (see also <https://www.embopress.org/page/journal/14693178/authorguide#datadeposition>). Please note that the Data Availability Section is restricted to new primary data that are part of this study. * Note - All links should resolve to a page where the data can be accessed. *

10) Regarding data quantification (see Figure Legends:

<https://www.embopress.org/page/journal/14693178/authorguide#figureformat>)

12) All Materials and Methods need to be described in the main text using our 'Structured Methods' format, which is required for all research articles. According to this format, the Methods section includes a Reagents and Tools Table (listing key reagents, experimental models, software and relevant equipment and including their sources and relevant identifiers) followed by a Methods and Protocols section describing the methods using a step-by-step protocol format. The aim is to facilitate adoption of the methodologies across labs. More information on how to adhere to this format as well as a downloadable template (.docx) for the Reagents and Tools Table can be found in our author guidelines:

An example of a Method paper with Structured Methods can be found here: <https://www.embopress.org/doi/full/10.1038/s44320-024-00037-6#sec-4>

You are able to opt out of this by letting the editorial office know (emboreports@embo.org). If you do opt out, the Review

Process File link will point to the following statement: "No Review Process File is available with this article, as the authors have chosen not to make the review process public in this case."

I look forward to seeing a revised form of your manuscript when it is ready.

Yours sincerely,

Referee #1:

In this manuscript, the authors demonstrate that the tachykinin-expressing (Tac1) neurons in the medial amygdala (MeA) regulate anxiety levels. They found that these tachykinin-expressing neurons are activated when the mice transition from a presumably high-anxiety region to a relatively low-anxiety region. Additionally, using chemogenetic interrogations, they show that the activity of MeA^{Tac1} neurons is crucial for regulating anxiety. Those results are novel and convincing. However, the authors employed several projection-specific targeting methods to examine the upstream and downstream regions of the MeA involved in anxiety regulation. I have two major comments regarding these results: they may require more evidence to support their conclusions.

Major comment1:

The authors demonstrated that the upstream sources with axon innervations on MeA^{Tac1} neurons comprise the majority of GABAergic neurons in the VTA, which exhibited changes in activity when transitioning between different presumed anxious states. They aimed to prove that the VTA^{GABA}→MeA^{Tac1} pathway regulates anxiety. The results (lines 153-177, Fig. 4) can only demonstrate that VTA-innervated MeA^{Tac1} neurons play a regulatory role in anxiety but cannot prove that this regulatory process is initiated by the VTA. This is because the VTA-innervated MeA^{Tac1} neurons may also receive innervation from other regions involved in regulating anxiety. To prove that the VTA^{GABA}→MeA^{Tac1} pathway is involved in anxiety regulation, the authors need to conduct more specific interrogation experiments on VTA^{GABA} neurons to verify their hypothesis. For instance, they could express hM4Di in the MeA-projecting neurons of the VTA and then perform a local infusion of CNO in the MeA. This approach would help to determine whether blocking the synaptic transmission of the VTA^{GABA}→MeA^{Tac1} pathway mimics the condition of relieving the VTA-mediated inhibition on MeA, thereby regulating anxiety levels. Additionally, they could combine this with hM4Di expression in MeA^{Tac1} neurons and carry out the same local infusion of CNO in the MeA to verify the upstream-downstream relationship of the VTA^{GABA}→MeA^{Tac1} pathway in anxiety regulation.

Major comment2:

The authors demonstrated that the downstream target of MeA^{Tac1} neurons is the VMHvl. They aimed to prove that MeA^{Tac1}→VMHvl projections inhibit anxiety. However, their results only indicate that VMHvl-projecting MeA^{Tac1} neurons are necessary to modulate anxiety. In lines 229-241 (Fig. 6), they aim to prove how the VTA^{GABA}→MeA^{Tac1}→VMHvl circuits modulate anxiety by blocking GABAergic transmission within the VMHvl when the VTA-innervated MeA^{Tac1} neurons were activated. However, by using bicuculline to block GABAergic transmission, they also inhibited GABAergic signals not initiated by MeA^{Tac1} input. The increased anxiety observed following local infusion of bicuculline in VMHvl may stem from the inhibition of GABAergic transmissions beyond MeA^{Tac1} input. If the authors wish to demonstrate that the VTA^{GABA}→MeA^{Tac1}→VMHvl circuits modulate anxiety, they need to specifically manipulate the synaptic transmission from VTA-innervated MeA^{Tac1} neurons to the downstream VMHvl while keeping other synaptic transmissions intact. For instance, they could employ the same projection-specific targeting methods illustrated in Fig. 6 to express hM4Di on VTA-innervated MeA^{Tac1} neurons and then perform local infusion of CNO within VMHvl to determine whether inhibiting this specific transmission can affect its ability to modulate anxiety.

I also have some minor comments about their manuscripts.

Minor comment1:

The sentence (lines 48-49), "The identification of the neural circuits in the MeA that are involved in the onset of anxiety..." may mislead readers into thinking that the MeA is responsible for initiating anxiety. Their results showed that the MeA did not increase activity during the transition from low to high anxious regions (Fig. 1H, 1I) but rather engaged in anxiety relief. Although they presented data indicating that inhibition of the MeA increased anxiety levels, this data may reflect a loss of anxiety-relieving function rather than a role in the onset of anxiety.

Minor comment2:

For all the calcium fiber photometry experiments conducted during the elevated plus maze task, the authors should clearly

specify how they define time zero for each type of event, including the exploration of open arms, the exploration of closed arms, transitions from open arms to closed arms, and transitions from closed arms to open arms.

Minor comment3:

I am confused about all the heatmaps of the calcium signals. The figure legend describes the color bars on the left, which represent different individual mice, but the unit on the heatmap figure is labeled "Bouts" rather than "Mice." I wonder if each row of the heatmap corresponds to the averaged calcium signal from a single mouse, and how many event bouts are used to calculate that averaged calcium signal. Alternatively, do the heatmap rows represent individual events from a single mouse?

Minor comment4:

I wonder if there is a typo in line 333: ferrule diameter of 2.5 "µm"?

Minor comment5:

Lines 354-356: Please provide more details about the rationale and how to estimate the instantaneous $\Delta F/F$ during the transitions between the open arms and closed arms of the maze.

Referee #2:

In this work, the authors provide a framework in which the Tac1-expressing neurons of the medial amygdala (MeA) contribute to high to low anxiety states transitions. Using circuit manipulation approaches the authors suggest that the circuit comprising inhibitory neurons of the midbrain projecting to Tac1 MeA that in turn send axons to the VMHvl nucleus of the hypothalamus are capable to limit anxiety states while the inhibition of this pathway leads to anxious phenotypes.

The authors used a combination of systems neuroscience approaches from photometry to chemogenetics. The manuscript remains quite difficult to follow and the description of experiments is somehow superficial. English proofreading might be appropriate in many instances. Several statements provided within the manuscript do not really reflect the data obtained. A major experimental problem is the use of CNO at the dose of 5mg/kg which remains very high especially knowing that D2 receptors are expressed throughout this area. Most of experimental setting in the field use now CNO doses at 1mg/kg or the more specific ligand CLZ. Another important aspect that remains elusive is the rationale for choosing the Tac1 neurons in this context. This should be better articulated within the introduction and discussion.

Based on these limitations and some of the elements described below this manuscript might be more appropriate for a more specialized journal.

Major concerns

1. In the fiber photometry experiments the level of the signal, its noise as well as the number of bouts are extremely variable. It is difficult to understand how data are obtained and quantified. In the panel K of Figure 1 at which time point the signal decays? The authors claim the transition from open to close is important however the fluorescent is not necessarily timelocked to such transition. The quantification is unclear as presented and statistics are missing (L and M panels). Overall the experimental details are few rendering difficult to grasp the data obtained.

2. The experiments using dreadd indicate overall time spent in open or close arms. However, the authors discuss the importance of Tac1 neurons in the transition from state to state. It is unclear how the two components relate one another. If the authors stress the transition concept, more and more precise experiments shall be performed. How these results tie in with photometry experiments?

3. In Figure 3 heatmaps are once again very different one another. In panel K the fluorescent mean is around 0 however the heatmap reports high level of fluorescence signal. This questions the experimental settings, the analysis performed and the description of the methods. In addition, this experiments loose the pathway specificity the authors are investigating.

4. The experiments with bicuculline are interesting. However, bicuculline is not labeled and is likely diffusing at higher rate in tissue. Can the authors rule out that the pharmacology has acted on neighboring structures?

5. The intersectional approaches used to dissect the pathway are puzzling. The dreadd expression in the end is always targeting MeA Tac1 neurons - how do these experiments provide better and refined pathway? It seems rather redundant with the initial findings while would have been important to study axonal activity within this circuit.

Minor

1. What is the meaning of sharing upon reasonable request? The authors might want to be more accessible when stating data sharing as journals and funding agencies are now supporting open science attitudes.

2. Some of the text should be revised as it does not fit with the literature for instance the paragraph on VTA function from line 126 below.

Referee #3:

The authors identified a novel circuit pathway, VTAGABA→MeATac1 →VMHvl, and its role in mediating anxiety behavior in mice. Overall, the findings are interesting, and the experiments are well-designed. However, some critical control experiments are missing, and the manuscript suffers a logical flaw. I have several major comments listed in chronological order:

1. The descriptions of anxiety in the summary and introduction are inaccurate. For example, anxiety is NOT a state in response to an immediate threat. And anxiety is not triggered by negative valence processing.
2. Although the authors showed using rabies that MeATac1-projecting cells in the VTA are mostly GABAergic, they will need to show that retroAAV also labels primarily GABAergic neurons in the VTA in Figure 3 before they can claim the recorded neuronal population is GABAergic. An alternative is to use VgatCre mice.
3. The authors showed that activation and inhibition of MeATac1 neurons bidirectionally control anxiety behavior. However, if VTA projection to the MeA is GABAergic, I would expect that manipulation of this projection produces the opposite behavioral effects. Yet, they showed that activation and inhibition of VTAGABA-MeATac1 produced a similar behavioral effect to MeATac1 neurons. How could it be possible?
4. The author claimed the trisynaptic projection in the paper, which is under-validated. The only experiment directly testing this is the last Figure. However, the authors cannot exclude the possibility that Bicuculline infusion in the VMHvl by itself can affect anxiety behavior regardless of VTA-MeATac1 projection.

Referee #1:

In this manuscript, the authors demonstrate that the tachykinin-expressing (Tac1) neurons in the medial amygdala (MeA) regulate anxiety levels. They found that these tachykinin-expressing neurons are activated when the mice transition from a presumably high-anxiety region to a relatively low-anxiety region. Additionally, using chemogenetic interrogations, they show that the activity of MeA^{Tac1} neurons is crucial for regulating anxiety. Those results are novel and convincing. However, the authors employed several projection-specific targeting methods to examine the upstream and downstream regions of the MeA involved in anxiety regulation. I have two major comments regarding these results: they may require more evidence to support their conclusions.

We wish to thank the reviewer for his insightful comments which we feel have substantially improved our manuscript. We believe we have addressed all of the major and minor concerns that were raised by the reviewer and have crafted a paper that is more rigorous in content and clearer in presentation.

Major comment1:

The authors demonstrated that the upstream sources with axon innervations on MeATac1 neurons comprise the majority of GABAergic neurons in the VTA, which exhibited changes in activity when transitioning between different presumed anxious states. They aimed to prove that the VTA^{GABA}→MeA^{Tac1} pathway regulates anxiety. The results (lines 153-177, Fig.

4) can only demonstrate that VTA-innervated MeA^{Tac1} neurons play a regulatory role in anxiety but cannot prove that this regulatory process is initiated by the VTA. This is because the VTA-innervated MeA^{Tac1} neurons may also receive innervation from other regions involved in regulating anxiety. To prove that the VTA^{GABA}→MeA^{Tac1} pathway is involved in anxiety regulation, the authors need to conduct more specific interrogation experiments on VTA^{GABA} neurons to verify their hypothesis. For instance, they could express hM4Di in the MeA-projecting neurons of the VTA and then perform a local infusion of CNO in the MeA. This approach would help to determine whether blocking the synaptic transmission of the VTA^{GABA}→MeA^{Tac1} pathway mimics the condition of relieving the VTA-mediated inhibition on MeA, thereby regulating anxiety levels. Additionally, they could combine this with hM4Di expression in MeA^{Tac1} neurons and carry out the same local infusion of CNO in the MeA to verify the upstream-downstream relationship of the VTA^{GABA}→MeA^{Tac1} pathway in anxiety regulation.

Response: We have performed additional behaviors experiments as the reviewer suggested (Figure S4).

“In the following experiments, C57BL/6J mice were bilaterally injected with AAV-DIO-mCherry (control) or AAV-DIO-hM4Di-mCherry virus into the VTA and a retrograde AAV-Retro-CAG-Cre virus into the MeA (Figure S4A, B). We found that inactivation of MeA-projecting VTA neuron activity through CNO (5 μM in 0.3 μl) treatment in C57BL/6J mice inhibited anxiety-like behaviors in mice (Figure S4C-H).” (line 168-173)

Figure for referees not shown.

In the following experiments, C57BL/6J mice were bilaterally injected with AAV-DIO-mCherry (control) or AAV-DIO-hM3Dq-mCherry virus into the VTA and a retrograde AAV-Retro-CAG-Cre virus into the MeA (**Figure S4I, J**). We found that activation of MeA-projecting VTA neuron activity through CNO (5 μ M in 0.3 μ l) treatment in C57BL/6J mice promoted anxiety-like behaviors in mice (**Figure S4K-P**). (line 183-188)

Major comment2:

The authors demonstrated that the downstream target of MeA^{Tac1} neurons is the VMHvl. They aimed to prove that MeA^{Tac1}→VMHvl projections inhibit anxiety. However, their results only indicate that VMHvl-projecting MeATac1 neurons are necessary to modulate anxiety. In lines 229-241 (Fig. 6), they aim to prove how the VTA^{GABA}→MeA^{Tac1}→VMHvl circuits modulate anxiety by blocking GABAergic transmission within the VMHvl when the VTA-innervated MeA^{Tac1} neurons were activated. However, by using bicuculline to block GABAergic transmission, they also inhibited GABAergic signals not initiated by

MeA^{Tac1} input. The increased anxiety observed following local infusion of bicuculline in VMHvl may stem from the inhibition of GABAergic transmissions beyond MeA^{Tac1} input. If the authors wish to demonstrate that the VTA^{GABA}→MeA^{Tac1}→VMHvl circuits modulate anxiety, they need to specifically manipulate the synaptic transmission from VTA-innervated MeA^{Tac1} neurons to the downstream VMHvl while keeping other synaptic transmissions intact. For instance, they could employ the same projection-specific targeting methods illustrated in Fig. 6 to express hM4Di on VTA-innervated MeA^{Tac1} neurons and then perform local infusion of CNO within VMHvl to determine whether inhibiting this specific transmission can affect its ability to modulate anxiety.

Response: We have performed additional behaviors experiments as the reviewer suggested.

Figure for referees not shown.

We next investigated whether the $VTA^{GABA} \rightarrow MeA^{Tac1} \rightarrow VMHvl$ circuit regulates anxiety through a combination of region-specific chemogenetic manipulations. We unilaterally injected AAV-fDIO-hM4Di-mCherry into the MeA and the anterograde virus AAV1-DIO-flpo into the VTA of *Tac1-Cre* mice. A cannula was then implanted into the VMHvl, through which CNO (5 μ M in 0.3 μ l) or saline was infused (**Fig. 6A, B**). After four weeks, behavioral assays were conducted to assess anxiety. CNO administration resulted in a significant reduction in the time spent exploring the center of the arena in the OFT (**Fig. 6C-E**). Similarly, in the EPM and LDB tests, mice infused with CNO spent less time in the open arms and light box (**Fig. 6F-H**). (line 227-236)

I also have some minor comments about their manuscripts.

Minor comment1:

The sentence (lines 48-49), "The identification of the neural circuits in the MeA that are involved in the onset of anxiety..." may mislead readers into thinking that the MeA is responsible for initiating anxiety. Their results showed that the MeA did not increase activity during the transition from low to high anxious regions (Fig. 1H, 1I) but rather engaged in anxiety relief. Although they presented data indicating that inhibition of the MeA increased anxiety levels, this data may reflect a loss of anxiety-relieving function rather than a role in the onset of anxiety.

Response: Thank you for pointing this out. We have rewritten this sentence as “The identification of the neural circuits in the MeA that are involved in the regulation of anxiety can enhance our understanding of the mechanisms in the brain underlying anxiety-related behavior.” (line 45-46)

Minor comment2:

For all the calcium fiber photometry experiments conducted during the elevated plus maze task, the authors should clearly specify how they define time zero for each type of event, including the exploration of open arms, the exploration of closed arms, transitions from open arms to closed arms, and transitions from closed arms to open arms.

Response: As the reviewer suggested, we have included the definition of time zero for each type of event from Figure 1 and 3 in the figure legends.

Minor comment3:

I am confused about all the heatmaps of the calcium signals. The figure legend describes the color bars on the left, which represent different individual mice, but the unit on the heatmap figure is labeled "Bouts" rather than "Mice." I wonder if each row of the heatmap corresponds to the averaged calcium signal from a single mouse, and how many event bouts are used to calculate that averaged calcium signal. Alternatively, do the heatmap rows represent individual events from a single mouse?

Response: Thank you for pointing this out. We have included new label and legends for all the heatmaps of the calcium signals.

Figure for referees not shown.

Figure for referees not shown.

Minor comment4:

I wonder if there is a typo in line 333: ferrule diameter of 2.5 "µm"?

Response: Thank you for pointing this out. We have rewritten this sentence as "A fiberoptic cannula (ferrule: Φ 2.5; fiber: 200-µm OD, 0.37 NA, 5.5 mm long) was inserted into the MeA." (line 355)

Minor comment5:

Lines 354-356: Please provide more details about the rationale and how to estimate the instantaneous $\Delta F/F$ during the transitions between the open arms and closed arms of the maze.

Response: We have included details about calcium imaging in the methods. "The data processing included correcting the raw fluorescence data (F) with the airPLS algorithm ($\lambda = 8$), followed by segmenting and aligning the data with the onset of behavioral events within individual trials or sessions. We calculated the relative fluorescence change values ($\Delta F/F$) as $(F-F_0)/(F_0-Voffset)$, where F_0 represents the baseline fluorescence signal (averaged over the -2 s to 0 s time window prior to a trigger event), and $Voffset$ is the fluorescence signal that was recorded prior to connecting the cannula to the optical fiber above the MeA. The resulting $\Delta F/F$ values were visualized as heatmaps or as average plots, with shaded areas indicating the standard error of the mean (SEM). During the elevated plus maze test, we estimated the instantaneous $\Delta F/F$ during transitions between the open arms and closed arms of the maze.

While the mice explore the elevated plus maze, the EV recording software will simultaneously record the mouse movement trajectories captured by EthoVision XT 11 and the calcium signal changes recorded by the dual-channel

software, ensuring that the occurrence of mouse events corresponds with the fluctuations in calcium signals. The time points when the mice just enter the open arm or the closed arm will be marked. All events are marked as starting from 0 seconds, and calcium signal analysis is conducted within a 10-second window before and after the event. The averaged calcium signal data for each mouse are directly calculated by MATLAB 2017b as the Peak and AUC.” (line 367-387)

Referee #2:

In this work, the authors provide a framework in which the Tac1-expressing neurons of the medial amygdala (MeA) contribute to high to low anxiety states transitions. Using circuit manipulation approaches the authors suggest that the circuit comprising inhibitory neurons of the midbrain projecting to Tac1 MeA that in turn send axons to the VMHvl nucleus of the hypothalamus are capable to limit anxiety states while the inhibition of this pathway leads to anxious phenotypes.

The authors used a combination of systems neuroscience approaches from photometry to chemogenetics. The manuscript remains quite difficult to follow and the description of experiments is somehow superficial. English proofreading might be appropriate in many instances. Several statements provided within the manuscript do not really reflect the data obtained. A major experimental problem is the use of CNO at the dose of 5mg/kg which remains very high especially knowing that D2 receptors are expressed throughout this area. Most of experimental setting in the field use now CNO doses at 1mg/kg or the more specific ligand CLZ. Another important aspect that remains elusive is the rationale for choosing the Tac1 neurons in this context. This should be better articulated within the introduction and discussion.

Based on these limitations and some of the elements described below this manuscript might be more appropriate for a more specialized journal.

We wish to thank the reviewer for his insightful comments which we feel have

substantially improved our manuscript. We believe we have addressed all of the major and minor concerns that were raised by the reviewer and have crafted a paper that is more rigorous in content and clearer in presentation.

Major concerns

1. In the fiber photometry experiments the level of the signal, its noise as well as the number of bouts are extremely variable. It is difficult to understand how data are obtained and quantified. In the panel K of Figure 1 at which time point the signal decays? The authors claim the transition from open to close is important however the fluorescent is not necessarily timelocked to such transition. The quantification is unclear as presented and statistics are missing (L and M panels). Overall the experimental details are few rendering difficult to grasp the data obtained.

Response: Thank you for pointing this out. We have included new label and legends for all the heatmaps of the calcium signals.

Figure for referees not shown.

Figure for referees not shown.

We have included details about calcium imaging in the methods.

“The data processing included correcting the raw fluorescence data (F) with the airPLS algorithm ($\lambda = 8$), followed by segmenting and aligning the data with the onset of behavioral events within individual trials or sessions. We calculated the relative fluorescence change values ($\Delta F/F$) as $(F-F_0)/(F_0-Voffset)$, where F_0 represents the baseline fluorescence signal (averaged over the -2 s to 0 s time window prior to a trigger event), and $Voffset$ is the fluorescence signal that was recorded prior to connecting the cannula to the optical fiber above the MeA. The resulting $\Delta F/F$ values were visualized as heatmaps or as average plots, with shaded areas indicating the standard error of the mean (SEM). During the elevated plus maze test, we estimated the instantaneous $\Delta F/F$ during transitions between the open arms and closed arms of the maze.

While the mice explore the elevated plus maze, the EV recording software will simultaneously record the mouse movement trajectories captured by EthoVision XT 11 and the calcium signal changes recorded by the dual-channel software, ensuring that the occurrence of mouse events corresponds with the fluctuations in calcium signals. The time points when the mice just enter the open arm or the closed arm will be marked. All events are marked as starting from 0 seconds, and calcium signal analysis is conducted within a 10-second window before and after the event. The averaged calcium signal data for each mouse are directly calculated by MATLAB 2017b as the Peak and AUC.” (line 367-387)

2. The experiments using dreadd indicate overall time spent in open or close arms. However, the authors discuss the importance of Tac1 neurons in the transition from state to state. It is unclear how the two components relate one another. If the authors stress the transition concept, more and more precise experiments shall be performed. How these results tie in with photometry experiments?

Response: We have rewritten the results as follows

'During the exploration of the elevated plus maze, when the mice move from the open arms to the closed arms, leading to a decrease in anxiety levels. Concurrently, the activity of Tac1 neurons increases, aligning with previous research findings²⁶. This suggests that Tac1 neurons in the MeA may play a role in regulating anxiety-like behavior in mice.'" (line 88-92)

26. Jimenez, J.C., Su, K., Goldberg, A.R., Luna, V.M., Biane, J.S., Ordek, G., Zhou, P., Ong, S.K., Wright, M.A., Zweifel, L., et al. (2018). Anxiety Cells in a Hippocampal-Hypothalamic Circuit. *Neuron* 97, 670-683 e676. 10.1016/j.neuron.2018.01.016.

3. In Figure 3 heatmaps are once again very different one another. In panel K the fluorescent mean is around 0 however the heatmap reports high level of fluorescence signal. This questions the experimental settings, the analysis performed and the description of the methods. In addition, this experiments loose the pathway specificity the authors are investigating.

Response: In the elevated plus maze experiments, as mice move from the open arms to the closed arms, they pass through a middle zone, with the time point of 0 indicating their entry into the closed arms. Before entering the closed arms, the external stimuli experienced by the mice have already changed, which is reflected in the altered activity of Tac1 neurons in the medial amygdala (MeA).

This represents a rising process that corresponds to the reduction in anxiety levels.

4. The experiments with bicuculline are interesting. However, bicuculline is not labeled and is likely diffusing at higher rate in tissue. Can the authors rule out that the pharmacology has acted on neighboring structures?

Response: The purpose of this experiment is to demonstrate that the $VTA^{GABA} \rightarrow MeA^{Tac1} \rightarrow VMHvl$ circuit regulates anxiety-like behavior. We do not have a reliable method to confirm whether the Bicuculline diffuse to other tissues. To verify whether the circuit is involved in the regulation of anxiety-like behavior, we conducted the following experiments.

Figure for referees not shown.

We next investigated whether the $VTA^{GABA} \rightarrow MeA^{Tac1} \rightarrow VMHvl$ circuit regulates anxiety through a combination of region-specific chemogenetic manipulations. We unilaterally injected AAV-fDIO-hM4Di-mCherry into the MeA and the anterograde virus AAV1-DIO-flpo into the VTA of *Tac1*-Cre mice. A cannula was then implanted into the VMHvl, through which CNO (5 μ M in 0.3 μ l) or saline was infused (**Fig. 6A, B**). After four weeks, behavioral assays were conducted to assess anxiety. CNO administration resulted in a significant reduction in the time spent exploring the center of the arena in the OFT (**Fig. 6C-E**). Similarly, in the EPM and LDB tests, mice infused with bicuculline spent less time in the open arms and light box (**Fig. 6F-H**). (line 227-236)

5. The intersectional approaches used to dissect the pathway are puzzling. The dreadd expression in the end is always targeting MeA *Tac1* neurons - how do these experiments provide better and refined pathway? It seems rather redundant with the initial findings while would have been important to study axonal activity within this circuit.

Response: Thank you for pointing this out. We have performed experiments as the reviewer suggested. (Figure S4)

“In the following experiments, C57BL/6J mice were bilaterally injected with AAV-DIO-mCherry (control) or AAV-DIO-hM4Di-mCherry virus into the VTA and a retrograde AAV-Retro-CAG-Cre virus into the MeA (**Figure S4A, B**). We found that inactivation of MeA-projecting VTA neuron activity through CNO (5

μM in 0.3 μl) treatment in C57BL/6J mice inhibited anxiety-like behaviors in mice (**Figure S4C-H**)." (line 168-173)

Figure for referees not shown.

In the following experiments, C57BL/6J mice were bilaterally injected with AAV-DIO-mCherry (control) or AAV-DIO-hM3Dq-mCherry virus into the VTA and a retrograde AAV-Retro-CAG-Cre virus into the MeA (**Figure S4I, J**). We found that activation of MeA-projecting VTA neuron activity through CNO (5 μ M in 0.3 μ l) treatment in C57BL/6J mice promoted anxiety-like behaviors in mice (**Figure S4K-P**). (line 183-188)

Minor

1. What is the meaning of sharing upon reasonable request? The authors might want to be more accessible when stating data sharing as journals and funding agencies are now supporting open science attitudes.

Response: Corrected

2. Some of the text should be revised as it does not fit with the literature for

instance the paragraph on VTA function from line 126 below.

Response: Corrected

Referee #3:

The authors identified a novel circuit pathway, VTAGABA→MeATac1 →VMHvl, and its role in mediating anxiety behavior in mice. Overall, the findings are interesting, and the experiments are well-designed. However, some critical control experiments are missing, and the manuscript suffers a logical flaw. I have several major comments listed in chronological order:

We wish to thank the reviewer for his insightful comments which we feel have substantially improved our manuscript. We believe we have addressed all of the major and minor concerns that were raised by the reviewer and have crafted a paper that is more rigorous in content and clearer in presentation.

1. The descriptions of anxiety in the summary and introduction are inaccurate. For example, anxiety is NOT a state in response to an immediate threat. And anxiety is not triggered by negative valence processing.

Response: We have rewritten the summary and introduction as the reviewer suggested

“Anxiety is an emotion characterized by worried thoughts and feelings of unease, often accompanied by physical symptoms such as sweating and dizziness. Unlike other negative emotions, the neural circuits underlying anxiety remain elusive. Herein, we reported that tachykinin-expressing (Tac1) neurons in the medial amygdala (MeA) respond to the transition from high anxiety to low anxiety states. The MeA^{Tac1} neurons regulate anxiety-like behaviors in mice bidirectionally. Moreover, we show that the GABAergic neurons in the ventral

tegmental area (VTA)^{GABA}→MeA^{Tac1}→ventrolateral part of the ventromedial hypothalamic nucleus (VMHvl) circuit contributes anxiety-like behaviors in mice. In summary, these findings reveal the circuit of Tac1 neurons in the MeA that mediates anxiety-like behaviors in mice.” (line 15-25)

“When the value judgment of an individual is negative, the individual experiences anxiety, which correspondingly triggers avoidance behaviors.”
(line 27-28)

2. Although the authors showed using rabies that MeATac1-projecting cells in the VTA are mostly GABAergic, they will need to show that retroAAV also labels primarily GABAergic neurons in the VTA in Figure 3 before they can claim the recorded neuronal population is GABAergic. An alternative is to use VgatCre mice.

Response: We have performed additional experiments as the reviewer suggested (figure 3B)

Figure for referees not shown.

3. The authors showed that activation and inhibition of MeATac1 neurons bidirectionally control anxiety behavior. However, if VTA projection to the MeA is GABAergic, I would expect that manipulation of this projection produces the opposite behavioral effects. Yet, they showed that activation and inhibition of VTAGABA-MeATac1 produced a similar behavioral effect to MeATac1 neurons. How could it be possible?

Response: We aimed to prove that the $VTA^{GABA} \rightarrow MeA^{Tac1}$ pathway regulates anxiety. The results demonstrated that VTA-innervated MeA^{Tac1} neurons play a regulatory role in anxiety. By the reason, this manipulation produced the same behavioral effects.

We have performed additional behaviors experiments as the reviewer suggested. (Figure S4)

“In the following experiments, C57BL/6J mice were bilaterally injected with AAV-DIO-mCherry (control) or AAV-DIO-hM4Di-mCherry virus into the VTA and a retrograde AAV-Retro-CAG-Cre virus into the MeA (**Figure S4A, B**). We found that inactivation of MeA-projecting VTA neuron activity through CNO (5 μ M in 0.3 μ l) treatment in C57BL/6J mice inhibited anxiety-like behaviors in mice (**Figure S4C-H**).” (line 168-173)

In the following experiments, C57BL/6J mice were bilaterally injected with AAV-DIO-mCherry (control) or AAV-DIO-hM3Dq-mCherry virus into the VTA and a retrograde AAV-Retro-CAG-Cre virus into the MeA (**Figure S4I, J**). We found that activation of MeA-projecting VTA neuron activity through CNO (5 μ M in 0.3 μ l) treatment in C57BL/6J mice promoted anxiety-like behaviors in mice (**Figure S4K-P**). (line 183-188)

Figure for referees not shown.

4. The author claimed the trisynaptic projection in the paper, which is under-validated. The only experiment directly testing this is the last Figure. However, the authors cannot exclude the possibility that Bicuculline infusion in the VMHvl by itself can affect anxiety behavior regardless of VTA-MeATac1 projection.

Response: Thank you for pointing this out. We have performed additional behaviors experiments as the reviewer suggested.

We next investigated whether the VTA^{GABA}→MeA^{Tac1}→VMHvl circuit regulates anxiety through a combination of region-specific chemogenetic manipulations. We unilaterally injected AAV-fDIO-hM4Di-mCherry into the MeA and the anterograde virus AAV1-DIO-flpo into the VTA of *Tac1*-Cre mice. A cannula was then implanted into the VMHvl, through which CNO (5 μM in 0.3 μl) or saline was infused (**Fig. 6A, B**). After four weeks, behavioral assays were conducted to assess anxiety. CNO administration resulted in a significant reduction in the time spent exploring the center of the arena in the OFT (**Fig.**

6C-E). Similarly, in the EPM and LDB tests, mice infused with bicuculline spent less time in the open arms and light box (**Fig. 6F-H**). (line 227-236)

Figure for referees not shown.

Dear Dr. He,

Thank you for the submission of your revised manuscript. We have now received the enclosed reports from the referees. As you will see, both referees 1 and 3 still have a few more comments that I would like you to address and incorporate before we can proceed with the official acceptance of your manuscript.

A few editorial requests will also need to be addressed:

- Please add up to 5 keywords to the ms file.
 - The Data Availability Section (DAS) needs to list a different sentence. If you have not deposited any data generated in this study in public databases please add this sentence to the DAS: This study includes no data deposited in external repositories. Please move the DAS to before the Acknowledgments.
 - Please correct the conflict of interest subheading to "Disclosure and Competing Interests Statement"; the separately uploaded Related Ms file is not needed.
 - Please remove the author credits from the ms file. All credits need to be entered during online ms submission.
 - The REFERENCE format needs to be corrected to the EMBO reports style: it needs to be alphabetical, not numerical; DOIs should only be used for preprints and datasets that have not been published yet
 - The author CHECKLIST has not selected responses to any of the questions in the checklist, please select responses to all questions and send us the completed checklist.
 - A callout for Figure 4OP is missing. Please add.
 - The PDF with suppl. figures needs to be updated to either 7 Expanded View (EV) figures that need to be individually uploaded and that will be integrated into the ms text online, or to an Appendix PDF file including a title page with a table of content with page numbers. Please see our guide to authors for more information. The figure titles and ms callouts need to be updated to either Figure EV1, etc... or Appendix Figure S1, etc..
The Suppl. Table can either be an EV table (Table EV1) or part of the Appendix file (Appendix Table S1). The table also needs a legend.
 - The Methods section should include a separate file called Reagents and Tools Table (listing key reagents, experimental models, software and relevant equipment and including their sources and relevant identifiers). A downloadable template (.docx) for the Reagents and Tools Table can be found in our author guidelines: <<https://www.embopress.org/page/journal/14693178/authorguide#manuscriptpreparation>>. Please upload the Reagents and Tools table with your final ms.
 - Please correct SUMMARY to Abstract.
 - During our routine image analysis of to be accepted ms we found the following issues:
 - *Image reuse between Figure 4N and Figure 5F - Not in legend
 - *Image reuse between Figure 6F and Appendix Fig S7F - Not in legend
 - *Microscopy images in Appendix Fig S3C are pixelated - this could have happened in conversion to jpg. Please request the authors to remake this figure with 16-bit TIFF captured image.Please explain the image reuses and remake the figure with the TIFF image.
- *Figure Legends - Comments*
- Please note that the exact p values are not provided in the legends of figures 2E, G, H, M, O, P; 4E, G, H, M, O, P; 5E, G, H, M, O, P; 6E, G, H, M, O, P. Please provide exact p-values as reasonable.
 - Please note that information related to n is missing in the legends of figures 1L, M; 2D, E, G, H, L, M, O, P; 4D, E, G, L, M, O; 5E, G, H, M, O, P; 6E, G, H, M, O, P. Please add.

I would like to suggest some minor changes to the abstract that needs to be written in present tense:

Anxiety is an emotion characterized by worried thoughts and feelings of unease, often accompanied by physical symptoms such as sweating and dizziness. Unlike other negative emotions, the neural circuits underlying anxiety are not well understood. Here

we report that tachykinin-expressing (Tac1) neurons in the medial amygdala (MeA) respond to the transition from high anxiety to low anxiety states. The MeATac1 neurons regulate anxiety-like behaviors in mice bidirectionally. We also show that GABAergic neurons in the ventral tegmental area (VTA)GABA→MeATac1→ventrolateral part of the ventromedial hypothalamic nucleus (VMHvl) circuit contribute to anxiety-like behavior in mice. Our findings reveal a circuit of Tac1 neurons in the MeA that mediates anxiety-like behaviors in mice.

EMBO press papers are accompanied online by A) a short (1-2 sentences) summary of the findings and their significance, B) 2-3 bullet points highlighting key results and C) a synopsis image that is exactly 550 pixels wide and 200-600 pixels high (the height is variable). The synopsis image should provide a sketch of the major findings, like a graphical abstract. Please note that text needs to be readable at the final size. Please send us this information along with the final manuscript.

Referee #1:

In this revised manuscript, the authors have adequately addressed the concerns I raised previously. Notably, they have improved the presentation of the figures and added methodological details regarding calcium fiber photometry, which strengthens their central conclusion about the role of MeA^{Tac1} neurons in regulating anxiety. Additionally, the authors now provide more substantial evidence for the upstream and downstream circuits involved in MeA^{Tac1}-mediated anxiety regulation. I have no major concerns about the revised manuscript; however, I do have two minor comments.

1. Lines 148- 151: There appears to be a discrepancy in the neuronal identity described in this section. Those neurons in the Figure legend are VTA^{GABA} neurons, but in the main text, they are referred to as MeA^{Tac1} neurons.

2. In Figures 6I-J, the authors aim to demonstrate that GABAergic transmission from MeA^{Tac1} to VMHvl modulates anxiety levels. They conducted an experiment involving the local infusion of bicuculline into the VMHvl while chemogenetically activating MeA^{Tac1} axonal activities. Although their results indicated that the GABAergic transmission in VMHvl is critical in regulating anxiety, those GABA actions may not be solely derived from MeA^{Tac1} neurons. Alternative sources-such as local GABAergic microcircuits within the VMHvl or other GABAergic afferents-may also contribute to the anxiolytic effects. They should at least discuss those possibilities in the discussion or perform additional control experiments to strengthen their conclusion. For example, they could replicate the experimental setup from Figure 6 to express hM4Di or hM3Dq in MeA^{Tac1} neurons, combined with local-VMHvl infusions comparing saline, saline + CNO, and bicuculline + CNO conditions.

Referee #2:

The authors have added a series of experiments in the present resubmission answering in large part to the concerns. The manuscript has been revised accordingly. This contribution will be interesting in the field.

Referee #3:

I appreciate the authors' efforts in addressing my previous comments. Overall, the manuscript can be considered for publication. My remaining issue is that the authors did not provide any quantification between GCaMP and GAD67 in Figure 3B. One high-resolution microscopic image does not sufficiently address my concern.

A few editorial requests will also need to be addressed:

Please add up to 5 keywords to the ms file.

Response: Done

- The Data Availability Section (DAS) needs to list a different sentence. If you have not deposited any data generated in this study in public databases please add this sentence to the DAS: This study includes no data deposited in external repositories. Please move the DAS to before the Acknowledgments.

Response: Corrected.

Please correct the conflict of interest subheading to "Disclosure and Competing Interests Statement"; the separately uploaded Related Ms file is not needed.

Response: Corrected.

Please remove the author credits from the ms file. All credits need to be entered during online ms submission.

Response: Corrected.

The REFERENCE format needs to be corrected to the EMBO reports style: it needs to be alphabetical, not numerical; DOIs should only be used for preprints and datasets that have not been published yet

Response: Corrected.

The author CHECKLIST has not selected responses to any of the questions in the checklist, please select responses to all questions and send us the completed checklist.

Response: Done.

A callout for Figure 4OP is missing. Please add.

Response: Corrected.

The PDF with suppl. figures needs to be updated to either 7 Expanded View (EV) figures that need to be individually uploaded and that will be integrated into the ms text online, or to an Appendix PDF file including a title page with a table of content with page numbers. Please see our guide to authors for more information. The figure titles and ms callouts need to be updated to either Figure EV1, etc... or Appendix Figure S1, etc..

The Suppl. Table can either be an EV table (Table EV1) or part of the Appendix file (Appendix Table S1). The table also needs a legend.

Response: Done

The Methods section should include a separate file called Reagents and Tools Table (listing key reagents, experimental models, software and relevant equipment and including their sources and relevant identifiers). A downloadable template (.docx) for the Reagents and Tools Table can be found in our author

guidelines. Please upload the Reagents and Tools table with your final ms.

Response: Done.

Please correct SUMMARY to Abstract.

Response: Corrected.

During our routine image analysis of to be accepted ms we found the following issues:

*Image reuse between Figure 4N and Figure 5F - Not in legend

*Image reuse between Figure 6F and Appendix Fig S7F - Not in legend

*Microscopy images in Appendix Fig S3C are pixelated - this could have happened in conversion to jpg. Please request the authors to remake this figure with 16-bit TIFF captured image.

Please explain the image reuses and remake the figure with the TIFF image.

Response: We sincerely apologize for the oversight that led to the repeated use of images. We have thoroughly reviewed all the images and revised new figures with TIFF images

Figure Legends - Comments*

- Please note that the exact p values are not provided in the legends of figures 2E, G, H, M, O, P; 4E, G, H, M, O, P; 5E, G, H, M, O, P; 6E, G, H, M, O, P.

Please provide exact p-values as reasonable.

Response: Done.

- Please note that information related to n is missing in the legends of figures 1L, M; 2D, E, G, H, L, M, O, P; 4D, E, G, L, M, O; 5E, G, H, M, O, P; 6E, G, H, M, O, P. Please add.

Response: We have added n for figures 1L and M. These experiments (Figure 2D, E, G, H) were conducted using the same group of mice, so we placed the n-values following panel H. We applied the same strategy for other figures as well.

I would like to suggest some minor changes to the abstract that needs to be written in present tense:

Anxiety is an emotion characterized by worried thoughts and feelings of unease, often accompanied by physical symptoms such as sweating and dizziness. Unlike other negative emotions, the neural circuits underlying anxiety are not well understood. Here we report that tachykinin-expressing (Tac1) neurons in the medial amygdala (MeA) respond to the transition from high anxiety to low anxiety states. The MeATac1 neurons regulate anxiety-like behaviors in mice bidirectionally. We also show that GABAergic neurons in the ventral tegmental area (VTA)GABA → MeATac1 → ventrolateral part of the ventromedial hypothalamic nucleus (VMHvl) circuit contribute to anxiety-like behavior in mice. Our findings reveal a circuit of Tac1 neurons in the MeA that mediates anxiety-

like behaviors in mice.

Response: Corrected.

EMBO press papers are accompanied online by A) a short (1-2 sentences) summary of the findings and their significance, B) 2-3 bullet points highlighting key results and C) a synopsis image that is exactly 550 pixels wide and 200-600 pixels high (the height is variable). The synopsis image should provide a sketch of the major findings, like a graphical abstract. Please note that text needs to be readable at the final size. Please send us this information along with the final manuscript.

Response: Done

Referee #1:

In this revised manuscript, the authors have adequately addressed the concerns I raised previously. Notably, they have improved the presentation of the figures and added methodological details regarding calcium fiber photometry, which strengthens their central conclusion about the role of MeA^{Tac1} neurons in regulating anxiety. Additionally, the authors now provide more substantial evidence for the upstream and downstream circuits involved in MeA^{Tac1}-mediated anxiety regulation. I have no major concerns about the revised manuscript; however, I do have two minor comments.

We wish to thank the reviewer for his insightful comments which we feel have

substantially improved our manuscript.

1. Lines 148- 151: There appears to be a discrepancy in the neuronal identity described in this section. Those neurons in the Figure legend are VTA^{GABA} neurons, but in the main text, they are referred to as MeA^{Tac1} neurons.

Response: Corrected.

2. In Figures 6I-J, the authors aim to demonstrate that GABAergic transmission from MeA^{Tac1} to VMHvl modulates anxiety levels. They conducted an experiment involving the local infusion of bicuculline into the VMHvl while chemogenetically activating MeA^{Tac1} axonal activities. Although their results indicated that the GABAergic transmission in VMHvl is critical in regulating anxiety, those GABA actions may not be solely derived from MeA^{Tac1} neurons. Alternative sources-such as local GABAergic microcircuits within the VMHvl or other GABAergic afferents-may also contribute to the anxiolytic effects. They should at least discuss those possibilities in the discussion or perform additional control experiments to strengthen their conclusion. For example, they could replicate the experimental setup from Figure 6 to express hM4Di or hM3Dq in MeA^{Tac1} neurons, combined with local-VMHvl infusions comparing saline, saline + CNO, and bicuculline + CNO conditions.

Response: We have rewritten the discussion part as the reviewer suggested.

“We observed that the introduction of GABA receptor antagonist, bicuculline into the downstream VMHvl region suppressed anxiety-like behaviors in mice. These findings suggest that GABA released from MeA^{Tac1} neurons may play a significant regulatory role in modulating anxiety-like behaviors. However, due to the limitations of our methodology, we cannot rule out the possibility that

GABA released from other brain regions project to the VMHvl also contributes to the regulation of anxiety-like behaviors in mice.” line (299-305)

Referee #2:

The authors have added a series of experiments in the present resubmission answering in large part to the concerns. The manuscript has been revised accordingly. This contribution will be interesting in the field.

We wish to thank the reviewer for his insightful comments which we feel have substantially improved our manuscript.

Referee #3:

I appreciate the authors' efforts in addressing my previous comments. Overall, the manuscript can be considered for publication. My remaining issue is that the authors did not provide any quantification between GCaMP and GAD67 in Figure 3B. One high-resolution microscopic image does not sufficiently address my concern.

We wish to thank the reviewer for his insightful comments which we feel have substantially improved our manuscript.

Referee #3:

We wish to thank the reviewer for his insightful comments which we feel have substantially improved our manuscript.

I appreciate the authors' efforts in addressing my previous comments. Overall, the manuscript can be considered for publication. My remaining issue is that the authors did not provide any quantification between GCaMP and GAD67 in Figure 3B. One high-resolution microscopic image does not sufficiently address my concern.

Response: We have revised the figure 3B as the reviewer suggested

Figure for referees not shown.

Dr. Zi-Xuan He
Northeast Normal University
school of life science
No.5268 Renmin Street
Jilin 130021
China

Dear Dr. He,

I am very pleased to accept your manuscript for publication in the next available issue of EMBO reports. Thank you for your contribution to our journal.
